# REFLECTIVE GAUSSIAN SPLATTING

**Yuxuan Yao**[1*], **Zixuan Zeng**[1*], **Chun Gu**[1], **Xiatian Zhu**[2†], **Li Zhang**[1†✉]

[1]School of Data Science, Fudan University    [2]University of Surrey

https://fudan-zvg.github.io/ref-gaussian

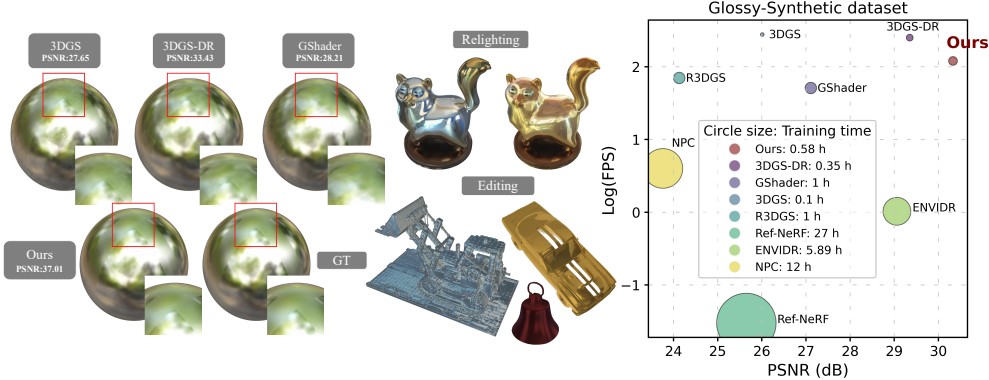

Figure 1: Our *Ref-Gaussian* achieves superior rendering quality in novel view synthesis while enjoying fast optimization (training time) and real-time rendering (FPS), and supporting various downstream applications such as *relighting* and *editing*. Please refer to the video in our *supplementary material* for a more comprehensive and intuitive comparison.

## ABSTRACT

Novel view synthesis has experienced significant advancements owing to increasingly capable NeRF- and 3DGS-based methods. However, reflective object reconstruction remains challenging, lacking a proper solution to achieve real-time, high-quality rendering while accommodating inter-reflection. To fill this gap, we introduce a Reflective Gaussian splatting (**Ref-Gaussian**) framework characterized with two components: (I) *Physically based deferred rendering* that empowers the rendering equation with pixel-level material properties via formulating split-sum approximation; (II) *Gaussian-grounded inter-reflection* that realizes the desired inter-reflection function within a Gaussian splatting paradigm for the first time. To enhance geometry modeling, we further introduce material-aware normal propagation and an initial per-Gaussian shading stage, along with 2D Gaussian primitives. Extensive experiments on standard datasets demonstrate that *Ref-Gaussian* surpasses existing approaches in terms of quantitative metrics, visual quality, and compute efficiency. Further, we show that our method serves as a unified solution for both reflective and non-reflective scenes, going beyond the previous alternatives focusing on only reflective scenes. Also, we illustrate that *Ref-Gaussian* supports more applications such as relighting and editing.

## 1 INTRODUCTION

Tremendous progress has been recently made in 3D object reconstruction and novel view synthesis from multiview images. Neural radiance field (NeRF) (Mildenhall et al., 2021) represents 3D scenes with neural implicit fields, achieving high-quality, photo-realistic renderings through volume rendering. However, NeRF is expensive computationally for optimization and rendering, limiting

---

[†]Co-last authorship
[✉]Li Zhang (lizhangfd@fudan.edu.cn) is the corresponding author.

its use in real-time applications. To address this, 3D Gaussian splatting (3DGS) Kerbl et al. (2023) instead represents a scene using a set of 3D Gaussians. Combining rasterization and alpha-blending, it achieves both real-time and high-quality rendering.

However, most NeRF and 3DGS struggle to model *reflective surfaces* due to the intrinsic inability of capturing high-frequency specular components. To mitigate this, Ref-NeRF (Verbin et al., 2022) leverages integrated directional encoding to smooth interpolation of reflected radiance. But this method fails to decompose the environment illumination, making it less useful for example unable to relight. A couple of NeRF methods (Verbin et al., 2022; Liang et al., 2023) are challenged by computational cost and limited flexibility. Using a simplified shading functions on each Gaussian can achieve rapid convergence, but suffering substantial noise in modeling of geometry, material, and lighting (Jiang et al., 2024). To avoid this issue, performing the shading function at the pixel level (i.e., deferred shading) following alpha-blending to smooth out the gradients is helpful (Ye et al., 2024). Nonetheless, limiting to a simplified shading function makes it ineffective to model complex reflective objects with complex structures and surfaces and the inter-reflection phenomenon. To address the latter, RelightableGaussian (Gao et al., 2023) traces rays across Gaussians for visibility inferring at the cost of heavy noises and added computational overhead from Monte Carlo sampling.

To address the aforementioned issues all together, we propose a Reflective Gaussian splatting (***Ref-Gaussian***) framework, allowing for real-time high-quality rendering of reflective objects while taking into account the complicated inter-reflection effect. This is made possible by two key components: **(I)** *Physically based deferred rendering*, where we empower the rendering equation with pixel-level material properties (e.g., Bidirectional reflectance distribution function, or BRDF in short), and adopt the split-sum approximation to avoid the heavy compute with Monte Carlo sampling. **(II)** *Gaussian-grounded inter-reflection*, where we realize the intricate inter-reflection function with Gaussian splatting for the very first time, despite the challenge of extracting the mesh accurately. To enhance the modeling of geometry, we choose 2D Gaussian primitives as scene representation, while introducing an initial per-Gaussian shading stage (performing shading at the Gaussian level to provide good geometric initialization), as well as a material-aware normal propagation process (enlarging the scales of Gaussians with high metallic and low roughness).

We make these **contributions**: **(I)** We strive to realize real-time high-quality rendering of reflective objects with inter-reflection in 3D space. **(II)** We innovate a generic 3D reconstruction framework, Reflective Gaussian splatting (***Ref-Gaussian***), characterized with *physically based deferred rendering* and *Gaussian-grounded inter-reflection*. This approach is further enhanced by geometry-focused optimization including adopting 2D Gaussian primitives, material-aware normal propagation, and per-Gaussian shading initialization. **(III)** Extensive experiments demonstrate that *Ref-Gaussian* outperforms previous methods in terms of quantitative metrics, visual quality, and compute efficiency under both reflective and non-reflective scenes, whilst supporting downstream applications such as relighting and editing.

## 2 RELATED WORK

**Novel view synthesis** aims to generate new images of a scene from unseen viewpoints with a limited set of observed images. Neural radiance fields (NeRF) (Mildenhall et al., 2020) employ volumetric rendering with implicit neural representations to produce highly detailed novel views from input images. Subsequent works have focused on two main directions: improving rendering quality (Barron et al., 2021; 2022; 2023) and accelerating both training and rendering speeds (Sun et al., 2022; Fridovich-Keil et al., 2022; Müller et al., 2022; Chen et al., 2022). More recently, 3D Gaussian splatting (3DGS) (Kerbl et al., 2023) proposed using 3D Gaussian primitives and tile-based differentiable rasterizer, achieving superior rendering quality and efficiency over NeRF-based methods. Building on 3DGS, methods like 2DGS (Huang et al., 2024) and GOF (Yu et al., 2024) have further explored surface reconstruction with Gaussian splatting. In this paper, we adopt 2D Gaussian primitives, as proposed in 2DGS, to represent scenes for more accurate geometry reconstruction.

**Inverse rendering** seeks to decompose scenes into geometry, lighting, and materials, posing a significant challenge due to the high-dimensional complexity of light interaction. NeRF (Mildenhall et al., 2020) is commonly used as a 3D representation in inverse rendering due to its flexible raymarching technique. Several NeRF-based methods (Srinivasan et al., 2021; Boss et al., 2021a; Zhang et al., 2021; Boss et al., 2021b; Yao et al., 2022; Boss et al., 2022; Zhang et al., 2023; Liu et al., 2023; Jin et al., 2023) model light interaction using the explicit rendering equation with

BRDF. For instance, NeRO (Liu et al., 2023) divides the training process into two stages: the first stage employs split-sum approximation to obtain detailed geometry, while the second stage focuses on accurate sampling to recover lighting and materials. In contrast, Ref-NeRF (Verbin et al., 2022) does not decompose material and environment lighting but instead shades points using a simplified function, limiting its applicability in downstream tasks such as relighting and editing.

With the advent of 3D Gaussian splatting (Kerbl et al., 2023), recent methods (Jiang et al., 2024; Gao et al., 2023; Liang et al., 2024; Shi et al., 2023) have begun to address inverse rendering tasks using Gaussian primitives for much faster rendering speed than NeRF methods. GShader (Jiang et al., 2024) applies a simplified shading functions from rendering equation on each Gaussian. While it achieves rapid convergence, this approach also brings substantial noise in geometry, material, and lighting. In contrast, 3DGS-DR (Ye et al., 2024) employs deferred shading by applying a simplified shading function at the pixel level, stabilizing the optimization process by smoothing out normal gradients. As both GShader and 3DGS-DR cannot model the inter-reflection effects, Re-lightableGaussian (Gao et al., 2023) introduces an innovative Gaussian-based ray tracing technique for visibility precomputation. However, it applies Monte Carlo sampling to evaluate the rendering equation, which slows down the rendering speed, and the inaccurate visibility also reduces the quality of reconstruction. In this paper, inspired by 3DGS-DR, we adopt pixel-level deferred shading but perform the rendering function with BRDF properties to handle complex scenes. Furthermore, we propose performing ray tracing on the extracted mesh for visibility computation in the specular component of our rendering equation. Together with the split-sum approximation of the rendering equation, we model inter-reflection effects while maintaining fast rendering speeds.

## 3 METHOD

Our Reflective Gaussian splatting (*Ref-Gaussian*) is formulated in the Gaussian splatting framework. The key components of *Ref-Gaussian* include (1) physically based deferred rendering and (2) Gaussian-focused inter-reflection. An overview of our pipeline is shown in Figure 2. We start with the preliminary of Gaussian splatting.

In standard 3DGS (Kerbl et al., 2023), each Gaussian is modeled as an ellipsoid, characterized by a 3D position $\boldsymbol{p}_k$ and a covariance matrix $\Sigma$, defined as:

$$\mathcal{G}(\boldsymbol{p}) = \exp\left(-\frac{1}{2}(\boldsymbol{p} - \boldsymbol{p}_k)^{\mathrm{T}}\Sigma^{-1}(\boldsymbol{p} - \boldsymbol{p}_k)\right). \tag{1}$$

During the rendering process, 3D Gaussians are projected into 2D Gaussians in camera space. The transformed 2D covariance matrix is approximated as: $\Sigma' = JW\Sigma W^{\mathrm{T}}J^{\mathrm{T}}$, where $W$ is the view transformation matrix and $J$ is the Jacobian of the perspective projection. Each Gaussian is also assigned an opacity $o$ and a view-dependent color $\boldsymbol{c}_i$ represented by spherical harmonics. The final pixel color $\boldsymbol{C}$ is computed by blending Gaussians in front-to-back order based on their depth:

$$\boldsymbol{C} = \sum_{i=1}^{N} \boldsymbol{c}_i T_i \alpha_i, \quad \text{where} \quad T_i = \prod_{j=1}^{i-1}(1 - \alpha_j), \tag{2}$$

where $\alpha_i$ is calculated by multiplying the opacity $o_i$ with the Gaussian weight, which is determined using the 2D covariance matrix $\Sigma'_i$.

**2D Gaussian primitive** However, 3D Gaussians struggle to capture detailed geometry due to inherent multi-view inconsistencies. We hence leverage 2D Gaussian splatting (Huang et al., 2024) where view-consistent 2D oriented planar Gaussian disks are used as rendering primitives.

Each 2D Gaussian is defined by its position $\boldsymbol{p}$, tangential vectors $\boldsymbol{t}_u$ and $\boldsymbol{t}_v$, and scaling factors $(s_u, s_v)$ that control its shape and size. Its influence is given by:

$$\mathcal{G}(u, v) = \exp\left(-\frac{u^2 + v^2}{2}\right), \tag{3}$$

where $(u, v)$ are coordinates in the local tangent space. The ray-splat intersection $(u, v)$ in this space is obtained via:

$$\boldsymbol{x} = (xz, yz, z, 1)^{\mathrm{T}} = WH(u, v, 1, 1)^{\mathrm{T}}, \quad \text{where } H = \begin{bmatrix} s_u\boldsymbol{t_u} & s_v\boldsymbol{t_v} & \boldsymbol{0} & \boldsymbol{p} \\ 0 & 0 & 0 & 1 \end{bmatrix} \in \mathbb{R}^{4\times4}. \tag{4}$$

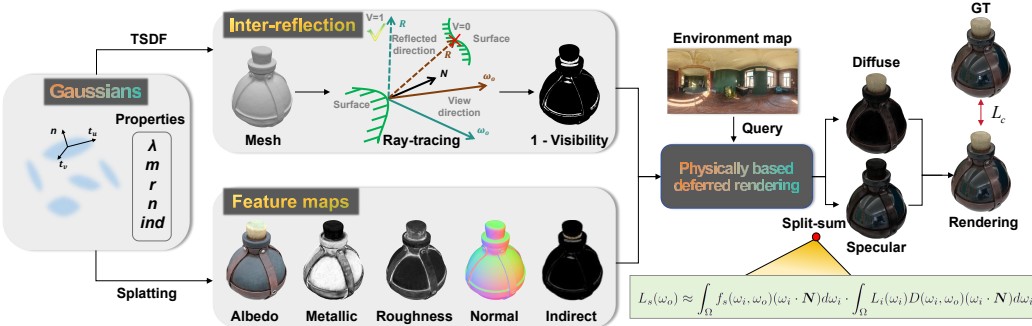

Figure 2: Overview of the **Ref-Gaussian** framework: First, we apply the splatting process to produce feature maps and perform ray-tracing on the extracted mesh to compute visibility for the specular term in the rendering equation. Next, we use the pixel-level feature maps to apply the rendering equation with split-sum approximation, yielding the final physically based rendering result.

Here, $\boldsymbol{x}$ represents the homogeneous ray passing through pixel $(x, y)$ and intersecting the 2D Gaussian disk at depth $z$. See more details of 2DGS in (Huang et al., 2024).

### 3.1 PHYSICALLY BASED DEFERRED RENDERING

Our physically based deferred rendering employs a simplified version of the Disney BRDF model (Burley and Studios, 2012). Specifically, each Gaussian is associated with a set of material-related properties, including albedo $\boldsymbol{\lambda} \in [0, 1]^3$, metallic $m \in [0, 1]$, and roughness $r \in [0, 1]$. The normal vector $\boldsymbol{n} \in [0, 1]^3$ can be derived from the tangential vectors of each 2D Gaussian using $\boldsymbol{n} = \boldsymbol{t_u} \times \boldsymbol{t_v}$. We deploy the rendering equation on pixel-level feature maps obtained via alpha-blending:

$$
\boldsymbol{X} = \sum_{i=1}^{N} \boldsymbol{x}_i \alpha_i \prod_{j=1}^{i-1} (1 - \alpha_j), \quad \text{where} \quad \boldsymbol{X} = \begin{bmatrix} \boldsymbol{\Lambda} \\ M \\ R \\ \boldsymbol{N} \end{bmatrix}, \quad \boldsymbol{x}_i = \begin{bmatrix} \lambda_i \\ m_i \\ r_i \\ \boldsymbol{n}_i \end{bmatrix}. \tag{5}
$$

Note, unlike shading directly on the Gaussians before alpha-blending (Jiang et al., 2024), our deferred shading treats alpha-blending as a smoothing filter. That allows to stabilize the optimization of features and finally produce more cohesive rendering results.

With the aggregated material maps, including albedo $\boldsymbol{\Lambda}$, metallic $M$, roughness $R$, and normal $\boldsymbol{N}$, the rendering equation is subsequently applied. The rendering equation expresses the outgoing radiance $L(\omega_o)$ in the direction $\omega_o$ as:

$$
L(\omega_o) = \int_{\Omega} L_i(\omega_i) f(\omega_i, \omega_o)(\omega_i \cdot \boldsymbol{N}) \, d\omega_i, \tag{6}
$$

where $f(\omega_i, \omega_o)$ denotes the bidirectional reflectance distribution function (BRDF). It represents the integral of reflected incident light over the hemisphere.

Since BRDF consists of both diffuse and specular terms, the integral can also be divided into two parts. The diffuse term is computed by querying the pre-integrated environment map using the normal $\boldsymbol{N}$ and multiplying with the material terms, while the specular term requires more complex reflection calculations to account for surface reflections. Here, let us focus on the specular component, where the specular term of the BRDF is given by:

$$
f_s(\omega_i, \omega_o) = \frac{D \, G \, F}{4(\omega_o \cdot \boldsymbol{N})(\omega_i \cdot \boldsymbol{N})}, \tag{7}
$$

where $D$, $F$, and $G$ represent the GGX normal distribution function, the Fresnel term, and the shadowing-masking term, respectively.

To compute the above integral term, a typical approach is Monte Carlo sampling as in (Gao et al., 2023). However, that is computationally expensive to render in real time. Instead, we adopt the split-sum approximation (Munkberg et al., 2022):

$$L_s(\omega_o) \approx \int_\Omega f_s(\omega_i, \omega_o)(\omega_i \cdot \boldsymbol{N})d\omega_i \cdot \int_\Omega L_i(\omega_i)D(\omega_i, \omega_o)(\omega_i \cdot \boldsymbol{N})d\omega_i, \tag{8}$$

where the first term depends solely on $(\omega_i \cdot \boldsymbol{N})$ and roughness $R$, allowing the results to be precomputed and stored in a 2D lookup texture map. The second term represents the integral of incident radiance over the specular lobe, and can be pre-integrated at the beginning of each training iteration with different roughness levels. That allows to efficiently employ a series of cubemaps to represent the environment lighting by performing trilinear interpolation using the reflected direction and the roughness as parameters.

## 3.2 GAUSSIAN GROUNDED INTER-REFLECTION

Inter-reflection plays a critical role in rendering reflective objects. To that end, we further enhance the specular light component by separately modeling the direct light $L_{\mathrm{dir}}$ (second term in Eq. 8) and indirect light $L_{\mathrm{ind}}$. Direct light $L_{\mathrm{dir}}$ denotes that whose reflection is not blocked by any scene elements, and the rest is defined as indirect light $L_{\mathrm{ind}}$.

Specifically, we approximate the visibility $V \in \{0, 1\}$ of incident light based on whether it is self-occluded along the reflected direction $\boldsymbol{R} = 2(w_o \cdot \boldsymbol{N})\boldsymbol{N} - w_o$. We represent the indirect light from occluded part as $L_{\mathrm{ind}}$:

$$L'_s(\omega_o) \approx \left( \int_\Omega f_s(\omega_i, \omega_o)(\omega_i \cdot \boldsymbol{N})d\omega_i \right) \cdot [L_{\mathrm{dir}} \cdot V + L_{\mathrm{ind}} \cdot (1 - V)]. \tag{9}$$

Intuitively, this indirect light component $L_{\mathrm{ind}}$ introduced here aims to effectively model the perturbation caused by occlusion in environmental lighting estimation.

Efficiently computing visibility along a ray across Gaussian primitives is non-trivial. Here, we propose ray tracing on extracted mesh on the fly. During optimization, we periodically extract the object's surface mesh using truncated signed distance function (TSDF) fusion. Once the mesh is constructed, we calculate the intersection points between rays and the surface using ray tracing, determining whether each pixel is occluded. For efficiency, we employ a bounding volume hierarchy (BVH) to accelerate the ray tracing process for visibility checks. As for direct lighting $L_{\mathrm{dir}}$, it just corresponds to the second term in Eq.8, handled as described in the previous section.

For the indirect lighting component, each Gaussian is assigned an additional view-dependent color $\boldsymbol{l}_{\mathrm{ind}}$, modeled by spherical harmonics. During the rendering process, $\boldsymbol{l}_{\mathrm{ind}}$ is evaluated in the reflected direction at the Gaussian level, and alpha blending is applied to aggregate the indirect lighting map as follows:

$$L_{\mathrm{ind}} = \sum_{i=1}^N \boldsymbol{l}_{\mathrm{ind}}\alpha_i \prod_{j=1}^{i-1}(1 - \alpha_j). \tag{10}$$

## 3.3 GEOMETRY FOCUSED MODEL OPTIMIZATION

To facilitate our physically based rendering, we further enhance the underlying geometry during optimization with the following designs.

**Initial stage with per-Gaussian shading**   As the cost for the benefit of smoothing gradients, pixel-level shading poses convergence challenges during the initial phase, because the shading function disturbs the gradient directions of the geometry. To address this, we propose a dedicated initial stage where the rendering equation is applied directly to each Gaussian primitive, using the material and geometry properties associated with it to compute the outgoing radiance. The outgoing radiance is then alpha-blended during rasterization to produce the final PBR rendering. This Gaussian-level shading design can help the gradients to be more effectively transferred back to the Gaussian primitives and facilitate eventually the optimization process.

**Material-aware normal propagation**   Positions with inaccurate normals often have difficulty capturing a significant specular component due to its intrinsic sensitivity to the reflected direction. We observe that in *Ref-Gaussian* there exists a strong positive correlation between normal accuracy and high metallic, low roughness properties (a typical situation where the specular component is significant). Under this insight, we propose periodically increasing the scale of 2D Gaussians with high

Table 1: Per-scene image quality comparison on synthesized test views. The intensity of the red color signifies a better result.

| Datasets | Shiny Blender (Verbin et al., 2022) | | | | | | Glossy Synthetic (Liu et al., 2023) | | | | | | | | Real (Verbin et al., 2022) | | |
|---|---|---|---|---|---|---|---|---|---|---|---|---|---|---|---|---|---|
| Scenes | ball | car | coffee | helmet | teapot | toaster | angel | bell | cat | horse | luyu | potion | tbell | teapot | garden | sedan | toycar |
| PSNR ↑ | | | | | | | | | | | | | | | | | |
| Ref-NeRF | 33.16 | 30.44 | 33.99 | 29.94 | 45.12 | 26.12 | 20.89 | 30.02 | 29.76 | 19.30 | 25.42 | 30.11 | 26.91 | 22.77 | 22.01 | 25.21 | 23.65 |
| ENVIDR | 41.02 | 27.81 | 30.57 | 32.71 | 42.62 | 26.03 | 29.02 | 30.88 | 31.04 | 25.99 | 28.03 | 32.11 | 28.64 | 26.77 | 21.47 | 24.61 | 22.92 |
| 3DGS | 27.65 | 27.26 | 32.30 | 28.22 | 45.71 | 20.99 | 24.49 | 25.11 | 31.36 | 24.63 | 26.97 | 30.16 | 23.88 | 21.51 | 21.75 | 26.03 | 23.78 |
| 2DGS | 25.97 | 26.38 | 32.31 | 27.42 | 44.97 | 20.42 | 26.95 | 24.79 | 30.65 | 25.18 | 26.89 | 29.50 | 23.28 | 21.29 | 22.53 | 26.23 | 23.70 |
| GShader | 30.99 | 27.96 | 32.39 | 28.32 | 45.86 | 26.28 | 25.08 | 28.07 | 31.81 | 26.56 | 27.18 | 30.09 | 24.48 | 23.58 | 21.74 | 24.89 | 23.76 |
| R3DG | 23.64 | 25.92 | 30.10 | 25.01 | 43.15 | 18.80 | 24.90 | 23.51 | 27.59 | 23.37 | 24.68 | 27.29 | 21.25 | 20.47 | 21.92 | 21.18 | 22.83 |
| 3DGS-DR | 33.43 | 30.48 | 34.53 | 31.44 | 47.04 | 26.76 | 29.07 | 30.60 | 32.59 | 26.17 | 28.96 | 32.65 | 29.03 | 25.77 | 21.82 | 26.32 | 23.83 |
| *Ref-Gaussian* | 37.01 | 31.04 | 34.63 | 32.32 | 47.16 | 28.05 | 30.38 | 32.86 | 33.01 | 27.05 | 30.04 | 33.07 | 29.84 | 26.68 | 22.97 | 26.60 | 24.27 |
| SSIM ↑ | | | | | | | | | | | | | | | | | |
| Ref-NeRF | 0.971 | 0.950 | 0.972 | 0.954 | 0.995 | 0.921 | 0.853 | 0.941 | 0.944 | 0.820 | 0.901 | 0.933 | 0.947 | 0.897 | 0.584 | 0.720 | 0.633 |
| ENVIDR | 0.997 | 0.943 | 0.962 | 0.987 | 0.995 | 0.922 | 0.934 | 0.954 | 0.965 | 0.925 | 0.931 | 0.960 | 0.947 | 0.957 | 0.561 | 0.707 | 0.549 |
| 3DGS | 0.937 | 0.931 | 0.972 | 0.951 | 0.996 | 0.894 | 0.792 | 0.908 | 0.959 | 0.797 | 0.916 | 0.938 | 0.900 | 0.881 | 0.571 | 0.771 | 0.637 |
| 2DGS | 0.934 | 0.930 | 0.972 | 0.953 | 0.997 | 0.892 | 0.918 | 0.911 | 0.958 | 0.909 | 0.918 | 0.939 | 0.902 | 0.886 | 0.609 | 0.778 | 0.597 |
| GShader | 0.966 | 0.932 | 0.971 | 0.951 | 0.996 | 0.929 | 0.914 | 0.919 | 0.961 | 0.933 | 0.914 | 0.936 | 0.898 | 0.901 | 0.576 | 0.728 | 0.637 |
| R3DG | 0.888 | 0.922 | 0.963 | 0.931 | 0.995 | 0.858 | 0.894 | 0.888 | 0.934 | 0.878 | 0.889 | 0.911 | 0.875 | 0.869 | 0.556 | 0.643 | 0.657 |
| 3DGS-DR | 0.979 | 0.963 | 0.976 | 0.971 | 0.997 | 0.942 | 0.942 | 0.959 | 0.973 | 0.933 | 0.943 | 0.959 | 0.958 | 0.942 | 0.581 | 0.773 | 0.639 |
| *Ref-Gaussian* | 0.981 | 0.964 | 0.976 | 0.971 | 0.998 | 0.948 | 0.954 | 0.969 | 0.973 | 0.944 | 0.952 | 0.963 | 0.962 | 0.947 | 0.617 | 0.777 | 0.660 |
| LPIPS ↓ | | | | | | | | | | | | | | | | | |
| Ref-NeRF | 0.166 | 0.050 | 0.082 | 0.086 | 0.012 | 0.083 | 0.144 | 0.102 | 0.104 | 0.155 | 0.098 | 0.084 | 0.114 | 0.098 | 0.251 | 0.234 | 0.231 |
| ENVIDR | 0.020 | 0.046 | 0.083 | 0.036 | 0.009 | 0.081 | 0.067 | 0.054 | 0.049 | 0.065 | 0.059 | 0.072 | 0.069 | 0.041 | 0.263 | 0.387 | 0.345 |
| 3DGS | 0.162 | 0.047 | 0.079 | 0.081 | 0.008 | 0.125 | 0.088 | 0.104 | 0.062 | 0.077 | 0.064 | 0.093 | 0.102 | 0.125 | 0.248 | 0.206 | 0.237 |
| 2DGS | 0.156 | 0.052 | 0.079 | 0.079 | 0.008 | 0.127 | 0.072 | 0.109 | 0.060 | 0.071 | 0.066 | 0.097 | 0.125 | 0.101 | 0.254 | 0.225 | 0.396 |
| GShader | 0.121 | 0.044 | 0.078 | 0.074 | 0.007 | 0.079 | 0.082 | 0.098 | 0.056 | 0.562 | 0.064 | 0.088 | 0.091 | 0.122 | 0.274 | 0.259 | 0.239 |
| R3DG | 0.214 | 0.058 | 0.090 | 0.125 | 0.013 | 0.170 | 0.085 | 0.125 | 0.089 | 0.081 | 0.080 | 0.117 | 0.156 | 0.115 | 0.354 | 0.380 | 0.312 |
| 3DGS-DR | 0.105 | 0.033 | 0.076 | 0.050 | 0.006 | 0.082 | 0.052 | 0.050 | 0.042 | 0.057 | 0.048 | 0.068 | 0.059 | 0.060 | 0.247 | 0.208 | 0.231 |
| *Ref-Gaussian* | 0.098 | 0.033 | 0.076 | 0.049 | 0.006 | 0.074 | 0.042 | 0.040 | 0.040 | 0.048 | 0.043 | 0.064 | 0.058 | 0.058 | 0.256 | 0.245 | 0.256 |

metallic and low roughness so that their normal information can be propagated to adjacent Gaussians for capturing more accurate geometry.

**Objective loss function**   We design a geometry focused objective function: $L = L_c + \lambda_n L_n + \lambda_{\text{smooth}} L_{\text{smooth}}$, where $L_c = (1 - \lambda)L_1 + \lambda L_{\text{D-SSIM}}$ is the RGB reconstruction loss where we set the balancing weight $\lambda = 0.2$ (Kerbl et al., 2023).

The second term $L_n = 1 - \tilde{N}^{\text{T}} N$ is the normal consistency loss. It encourages alignment of the Gaussians with the surface by minimizing the cosine difference between the rendered normal $N$, obtained via alpha-blending, and the surface normal $\tilde{N}$, derived from the depth map.

The third term $L_{\text{smooth}} = \|\nabla N\| \exp(-\|\nabla C_{gt}\|)$ is an edge-aware normal smoothness loss. It aims to regularize normal variation in regions of low texture.

## 4   EXPERIMENTS

To evaluate the performance of *Ref-Gaussian* in novel view synthesis, as well as geometry, materials, and lighting reconstruction, and downstream applications, we conduct a series of quantitative and qualitative experiments. Additionally, We provide a demo video in the *supplementary material*, showcasing dynamic renderings of *Ref-Gaussian* across two synthetic datasets, along with a comparative analysis.

**Datasets** We select two synthetic datasets Shiny Blender (Verbin et al., 2022) and Glossy Synthetic (Liu et al., 2023) for novel view synthesis of reflective objects, and Ref-Real dataset (Verbin et al., 2022) for real-world open scenes.

**Competitors** We compare several representative models in the field of reflective 3D reconstruction, including Ref-NeRF (Verbin et al., 2022), ENVIDR (Liang et al., 2023), 3DGS (Kerbl et al., 2023), GaussianShader (Jiang et al., 2024), RelightableGaussian (R3DG) (Gao et al., 2023), and 3DGS-DR (Ye et al., 2024). We also compared with more NeRF-based models like NeRO (Liu et al., 2023), NeuS (Wang et al., 2021) and NDE (Wu et al., 2024), see Tables 5 and 6 in appendix.

**Evaluation metrics** We use three standard metrics: PSNR, SSIM (Wang et al., 2004), and LPIPS (Zhang et al., 2018).

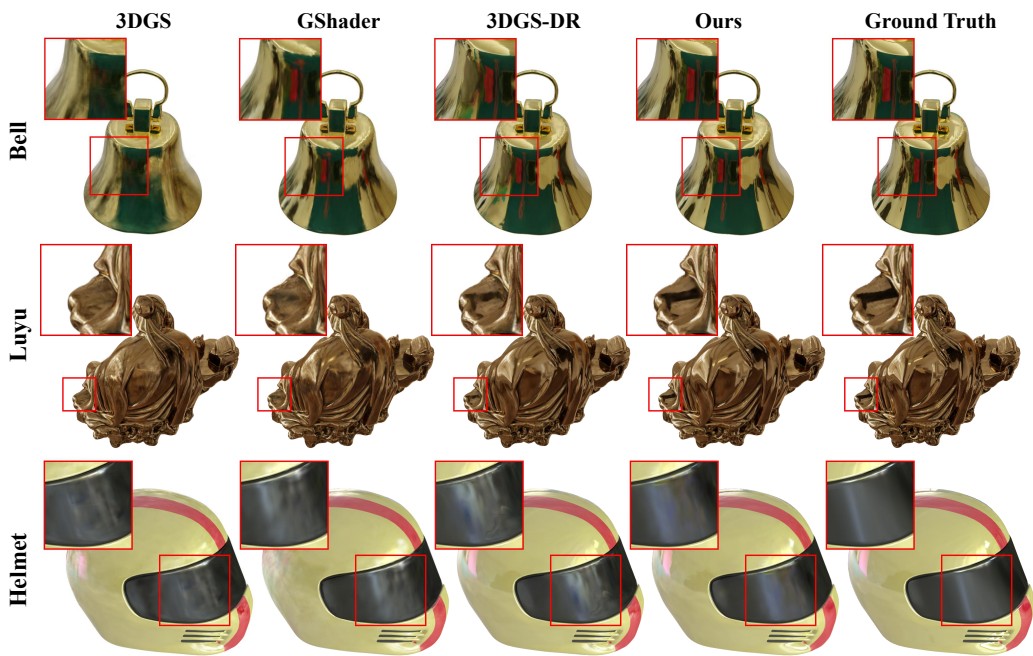

Figure 3: Qualitative comparisons on reflective scenes, including Bell, Luyu and Helmet.

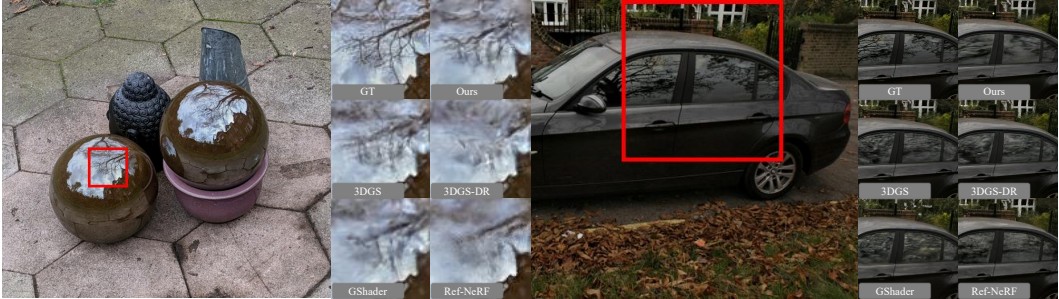

Figure 4: Qualitative comparisons on Ref-Real dataset (Verbin et al., 2022).

**Implementation details** Our training procedure consists of two stages. We utilize a per-Gaussian rendering for $18,000$ steps as an initial stage, followed by a deferred rendering stage training for about $40,000$ steps. We reset all color and material attributes before the second stage, retaining only the the geometry of Gaussians. The learning rates of the trainable material attributes (metallic, roughness, and albedo) are all set to $0.005$, while the learning rate of the environment map is set to $0.01$. The set of other basic trainable attributes for Gaussians like position and covariance is consistent with 2DGS (Huang et al., 2024). Then, our material-aware normal propagation is conducted to those Gaussians with metallic no less than $0.02$ and roughness no more than $0.1$. Also, we adopt the metallic initial value from 3DGS-DR (Ye et al., 2024) and set the initial roughness value to be $0.1$. During the implementation of physically based rendering, we discovered that spherical harmonics have a better fitting capability than the integrated diffuse lighting and we thereby use spherical harmonics to substitute it. For the loss function, $\lambda_n$ is set to $0.05$ and $\lambda_{\text{smooth}}$ is $1.0$. Additionally, we periodically extract object's surface mesh at 3000 step intervals. All experiments are conducted on a single NVIDIA A6000 GPU.

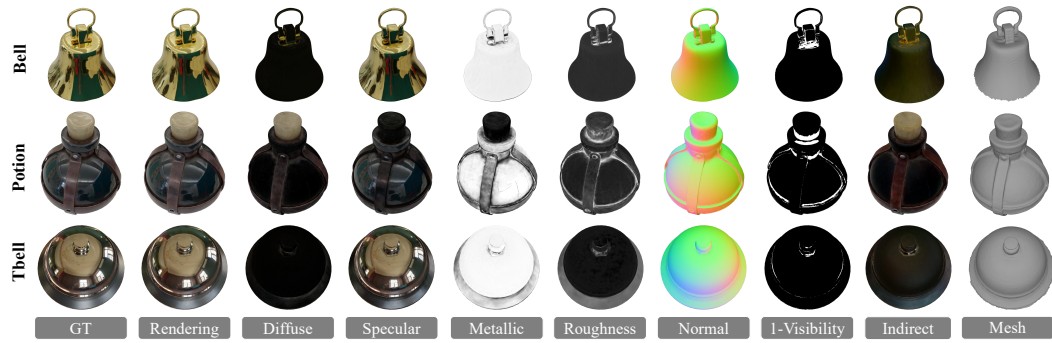

Figure 5: Inverse rendering with extracted mesh and indirect light. **Indirect**: Only consider indirect light as specular component when rendering.

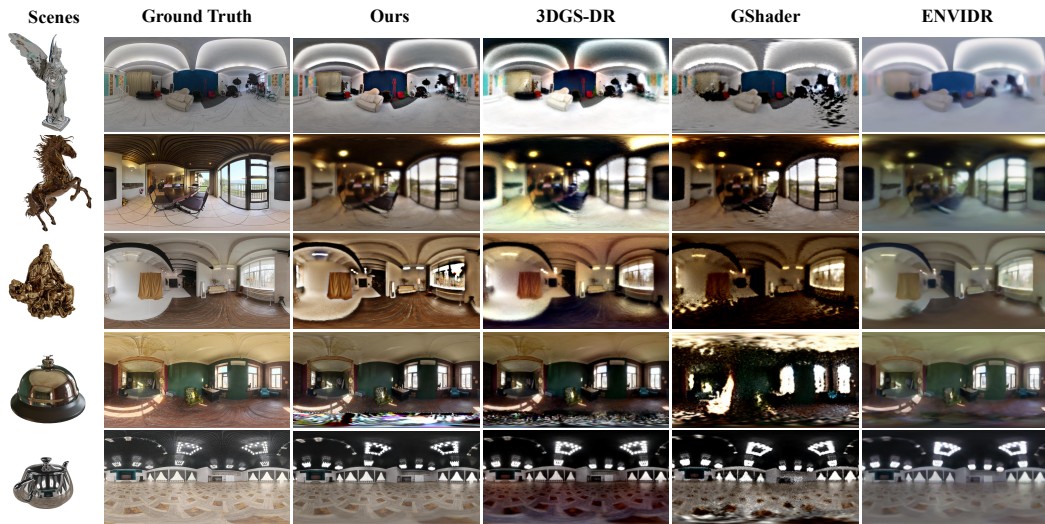

Figure 6: Qualitative comparisons of the estimated environment maps on Glossy Synthetic dataset (Liu et al., 2023).

## 4.1 COMPARISONS

**Novel view synthesis** In Table 1, we present the quantitative results of several competitors, along with *Ref-Gaussian*, for the novel view synthesis task across multiple datasets. *Ref-Gaussian* achieves state-of-the-art performance in most scenes, especially with significant improvements on glossy synthetic datasets, where reflective surfaces dominate. As shown in Figure 3, 3DGS (Kerbl et al., 2023) struggles to handle high-frequency, view-dependent effects, resulting in noticeable blurriness on reflective surfaces in multi-view scenarios, such as Bell. GShader (Jiang et al., 2024) introduces significant noise, leading to distortion. 3DGS-DR (Ye et al., 2024), which employs a simplified shading function, finds it challenging to accurately model complex reflective scenes, as observed in Luyu and the rough tinted visor on Helmet. In contrast, *Ref-Gaussian* effectively overcomes these challenges, offering precise environmental lighting modeling and robust handling of complex geometries and materials. Additionally, in Figure 4, we present a qualitative comparison of real-world scenes (Verbin et al., 2022). *Ref-Gaussian* provides more accurate normal maps and lighting, capturing subtle nuances on reflective surfaces. Specifically, the reflection phenomena on the garden sphere in the Garden and the car window in the Sedan are particularly noticeable. We demonstrate the more physically accurate reflection detail capture ability of *Ref-Gaussian* in Figure 4.

**Material, normal, and light estimation** In Table 2, we present a quantitative comparison against competitors in terms of normal maps, environment maps, and efficiency. *Ref-Gaussian* demonstrates

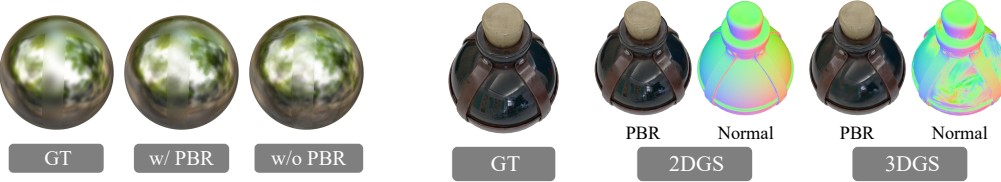

Figure 7: Ablation study on PBR.

Figure 8: Ablation study on usage of 2DGS.

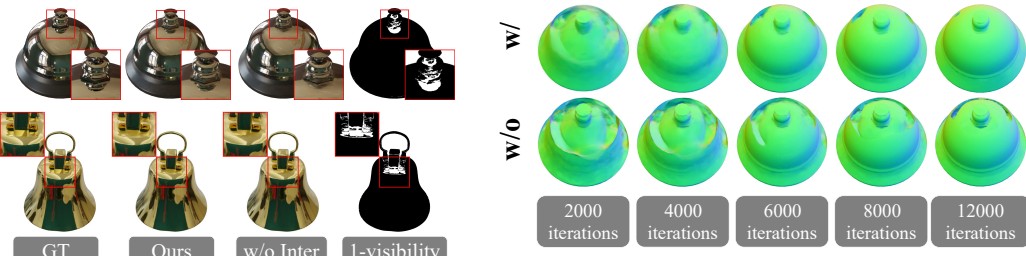

Figure 9: Ablation study on inter-reflection.

Figure 10: Effect of material-aware normal propagation on normal maps at various steps.

superior geometry reconstruction, as demonstrated by the high quality of the normal maps. This improvement is largely attributed to the carefully designed geometry optimization strategies discussed in Section 3.3. As shown in Figure 5, our normal map is smooth, as well as other decomposed material maps. Additionally, our visibility map is sharp and accurately highlights areas occluded in the reflected direction. *Ref-Gaussian* also achieves the best results among competitors in estimating environment lighting, which is evaluated after re-scaling to eliminate ambiguity. Figure 6 presents a qualitative comparison of the estimated environment maps. Compared to previous methods, our results more closely resemble the ground truth in terms of both detail and overall tone, while exhibiting the least amount of noise. This improvement can be attributed to the precise geometry and comprehensive modeling employed in *Ref-Gaussian*.

Furthermore, *Ref-Gaussian* excels in optimization speed, demonstrating rapid convergence, and supports real-time rendering. The average training time and frame-per-second (FPS) rendering performance are evaluated on the Shiny Blender dataset (Verbin et al., 2022).

## 4.2 ABLATION STUDY

We provide quantitative ablation studies for *Ref-Gaussian* below.

**Physically based rendering**     Assigning BRDF properties to Gaussians greatly enhances their ability to represent complex scenes. Compared to the simplified shading function using only one reflective direction vector to query the environment map as in 3DGS-DR (Ye et al., 2024), *Ref-Gaussian* demonstrates substantially better performance when learning scenes with material variations, such as the rough band on the sphere (Figure 7) and the visor on the helmet (Figure 3). Also, physically based rendering contributes to the geometry reconstruction, as verified by the results in Table 3.

**Gaussian-grounded inter-reflection**     As shown in Figure 9, the impact of our inter-reflection component is particularly evident on the head part of Tbell and Bell. With ray-traced visibility, we manage to reconstruct indirect lighting for novel view synthesis. Note that, inter-reflection is the minority in the glossy synthetic dataset. Its effect cannot be fully observed from Table 4.

**Deferred rendering**     We conduct physically based rendering at the per-Gaussian level and observe that the modeling of reflective details becomes noticeably more blurred. As shown in Table 4, without deferred rendering, all quality metrics of the rendered output drop significantly.

Table 2: Quantitative comparison on normal maps, environment maps, and efficiency, with all metrics averaged across datasets.

Table 3: Ablation studies on geometry optimization techniques.

| Model | Normal | | | Envmap | Efficiency | |
|---|---|---|---|---|---|---|
| | MAE↓ | SSIM↑ | LPIPS↓ | LPIPS↓ | Training time (h)↓ | FPS↑ |
| ENVIDR (Liang et al., 2023) | 2.74 | 0.948 | 0.0728 | 0.5099 | 5.84 | 1 |
| GShader (Jiang et al., 2024) | 7.00 | 0.931 | 0.1109 | 0.6145 | 0.48 | 28 |
| 3DGS-DR (Ye et al., 2024) | 2.62 | 0.947 | 0.0645 | 0.5290 | 0.35 | 251 |
| *Ref-Gaussian* | 2.15 | 0.954 | 0.0511 | 0.4431 | 0.58 | 122 |

| Model | MAE |
|---|---|
| *Ref-Gaussian* | 2.15 |
| w/o 2DGS | 4.45 |
| w/o Initial stage | 3.53 |
| w/o Material-aware | 3.86 |
| w/o PBR | 3.81 |

Table 4: Ablation studies on the components of *Ref-Gaussian*. w/o Material-aware: use original normal propagation following 3DGS-DR (Ye et al., 2024).

| Model | *Ref-Gaussian* | w/o PBR | w/o Inter-reflection | w/o Deferred rendering | w/o Initial stage | w/o Material-aware |
|---|---|---|---|---|---|---|
| **PSNR↑** | 30.33 | 29.84 | 30.14 | 28.85 | 29.94 | 29.68 |
| **SSIM↑** | 0.958 | 0.956 | 0.957 | 0.935 | 0.956 | 0.952 |
| **LPIPS↓** | 0.049 | 0.052 | 0.050 | 0.074 | 0.052 | 0.056 |

**2D Gaussian primitive** The results, as shown in Figure 8 , Table 3 and Table 7 in the appendix, emphasize the superiority of 2D Gaussian primitives over 3D alternative for more precise geometric representation. This is because 3D Gaussian conflicts with the thin nature of surfaces for its volumetric radiance representation. Notably, replacing 2DGS with 3DGS and keeping other components unchanged, Ref-Gaussian can still achieve outstanding performance across Shiny Blender (Verbin et al., 2022) and Glossy Synthetic (Liu et al., 2023) datasets, as shown in Table 7 in the appendix.

**Per-Gaussian shading initialization** The per-Gaussian physically based rendering used in the initial stage offers notable advantages in both the speed and accuracy of geometric convergence. When we replace the initial stage with an equal number of steps using physically based deferred rendering, the final results show a decline in rendering quality and normal reconstruction quality across various metrics.

**Material-aware normal propagation** The material-aware normal propagation accelerates the convergence of smooth surfaces. The quantitative comparisons presented in Tables 3 and 4 demonstrate its effectiveness in geometry reconstruction. As shown in Figure 10, smooth surfaces are prone to geometric collapse, but our material-aware normal propagation effectively addresses this issue.

### 4.3 APPLICATION TO RELIGHTING AND EDITING

*Ref-Gaussian* is a generic model, supporting many applications such as relighting and editing. For relighting, we model the diffuse term using integrated environment lighting, allowing to be modified. Similarly, we can modify the material properties of the Gaussians to edit the reconstructed scenes. We showcase the results in Figure 11 (appendix).

## 5 CONCLUSION

In this paper, we have introduced **Ref-Gaussian**, a novel Gaussian splatting framework designed for reconstruction of reflective objects/scenes in real time and high quality. This is achieved by formulating two integrated components: physically based deferred rendering and Gaussian grounded inter-reflection, along with geometry focused model optimisation. We also show the importance of choosing the basis representation model with 2DGS preferred over 3D alternative. Extensive experiments on multiple datasets demonstrate the superiority of *Ref-Gaussian* over existing methods in terms of quantitative metrics, visual quality, and efficiency. Also, we showcase the effect of *Ref-Gaussian* for other applications such as relighting and editing.

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

# A APPENDIX

## A.1 LIMITATIONS

As *Ref-Gaussian* is specifically designed for reconstructing reflective scenes, it might not excel for non-specular scenes. When modeling inter-reflections, we only consider the visibility in the reflection direction to approximate the overall visibility across the entire specular lobe. While this approach significantly reduces the computational complexity of rendering, it might introduce some noise in material estimation due to the inherent inaccuracy with the approximation.

## A.2 QUALITATIVE RESULTS OF RELIGHTING AND EDITING

We illustrate the results of both applications with realistic visual effects in Figure 11.

## A.3 FURTHER QUANTITATIVE COMPARISONS ON NON-SPECULAR DATASET

By design, *Ref-Gaussian* excel for both reflective and non-reflective scenes, **making it as a unified soluton for modeling a variety of scenes with varying reflection**. This is because, as described in Eq.9, inter-reflection is incorporated just as an additional element into the specular component. Consequently, in less reflective scenes, the influence of inter-reflection diminishes naturally and accordingly as the metallic attribute decreases. In such cases, the diffuse component instead takes over, including the capture of any (usually substantially weak) inter-reflection effect to achieve top performance. This behavior is evident in the decomposition results provided in Figure 5 and Figure 14 in the appendix. In contrast, previous reflection focused alternatives such as 3DGS-DR (Ye et al., 2024) are designed specifically only for reflective scenes, leading to narrowed applications as mentioned.

To support this claim, we have included Table 8 in the appendix, which presents a per-scene quantitative comparison among *Ref-Gaussian*, 3DGS-DR (Ye et al., 2024), and 3DGS (Kerbl et al., 2023) on the NeRF-Synthetic dataset (Mildenhall et al., 2020), predominantly composed of non-reflective scenes, for novel view synthesis. The results show that our *Ref-Gaussian* still excels over both alternatives, validating its generic and unified advantages.

## A.4 EXTRA BASELINES FOR NERF-BASED METHOD

We have further compared the average PSNR, SSIM, LPIPS, and FPS on the Glossy Blender dataset with NeRO (Liu et al., 2023) and NeuS (Wang et al., 2021) in Table 5, as well as with NDE (Wu et al., 2024) on the Shiny Blender dataset in Table 6. The quantitative results demonstrate that *Ref-Gaussian* significantly outperforms NeRO and remains comparable to NDE in rendering quality while achieving significantly better rendering speed and training efficiency.

Table 5: NVS quality of NeuS (Wang et al., 2021) and NeRO (Liu et al., 2023) on the Glossy-Blender dataset (Liu et al., 2023).

| Model | NeuS | NeRO | *Ref-Gaussian* |
|---|---|---|---|
| **PSNR↑** | 27.86 | 29.73 | 30.33 |
| **SSIM↑** | 0.878 | 0.904 | 0.958 |
| **LPIPS↓** | 0.375 | 0.324 | 0.049 |
| **FPS↑** | 0.02 | 0.11 | 122 |

Table 6: NVS quality of NDE (Wu et al., 2024) on the Shiny-Blender dataset (Verbin et al., 2022).

| Model | NDE | *Ref-Gaussian* |
|---|---|---|
| **PSNR↑** | 35.48 | 35.04 |
| **SSIM↑** | 0.976 | 0.973 |
| **LPIPS↓** | 0.027 | 0.056 |
| **FPS↑** | 66 | 122 |

Table 7: Per-scene PSNR comparison on synthesized test. w/o 2DGS: Using 3DGS as the representation of our *Ref-Gaussian* with the rest unchanged.

| Datasets | Shiny Blender (Verbin et al., 2022) | | | | | | Glossy Synthetic (Liu et al., 2023) | | | | | | | |
|---|---|---|---|---|---|---|---|---|---|---|---|---|---|---|
| Scenes | ball | car | coffee | helmet | teapot | toaster | angel | bell | cat | horse | luyu | potion | tbell | teapot |
| ENVIDR | 41.02 | 27.81 | 30.57 | 32.71 | 42.62 | 26.03 | 29.02 | 30.88 | 31.04 | 25.99 | 28.03 | 32.11 | 28.64 | 26.77 |
| 3DGS-DR | 33.43 | 30.48 | 34.53 | 31.44 | 47.04 | 26.76 | 29.07 | 30.60 | 32.59 | 26.17 | 28.96 | 32.65 | 29.03 | 25.77 |
| w/o 2DGS | 36.10 | 30.65 | 34.51 | 33.29 | 44.25 | 27.03 | 28.33 | 30.60 | 33.14 | 26.70 | 29.35 | 32.94 | 29.17 | 26.31 |
| *Ref-Gaussian* | 37.01 | 31.04 | 34.63 | 32.32 | 47.16 | 28.05 | 30.38 | 32.86 | 33.01 | 27.05 | 30.04 | 33.07 | 29.84 | 26.68 |

## A.5 QUALITATIVE RESULTS OF INDIRECT COMPONENT ON REF-REAL DATASET

We additionally provided richer qualitative results in Figure 12 where multiple objects provide rich inter-reflection. It can be observed that, in the task of novel view synthesis, the three components—diffuse, specular, and indirect light—complement each other effectively and together produce an excellent final rendering result. Specifically, based on the metallic property, the diffuse component primarily models non-reflective objects and ground details outside the stone sphere. The specular component focuses on the parts of the stone sphere where reflected light is not occluded by the ground or other objects, capturing fine reflective details, and the indirect component effectively captures the projections of the ground and adjacent objects. These decomposition results strongly demonstrate that *Ref-Gaussian*, even in complex real-world scenes, can accurately capture various details through its well-designed rendering approach and precise geometric reconstruction.

## A.6 QUANTITATIVE RESULTS OF COMPREHENSIVE 2DGS ABLATION STUDY

To explicitly showcase the effect of 2DGS representation and to more fairly and comprehensively demonstrate the superiority of *Ref-Gaussian* with other techniques, over other 3DGS-based methods, we replace the 2DGS representation of *Ref-Gaussian* with 3DGS while keeping the rest unchanged for comparison as shown in Table 7. The introduction of 2DGS has significantly enhanced the model's performance. And it is worth emphasizing that the *Ref-Gaussian* built on 3DGS, also excels in most scenarios of the Shiny Blender (Verbin et al., 2022) and Glossy Synthetic (Liu et al., 2023) datasets, further highlighting the inherent superiority of the rendering model.

## A.7 QUALITATIVE RESULTS OF GEOMETRY RECONSTRUCTION

We provide further illustration of our advantage in geometry reconstruction (Figure 13, 14). The qualitative comparison in Figure 13 demonstrates *Ref-Gaussian*'s comprehensive grasp of details over other advanced techniques (such as the tires in car and the water surface in coffee).

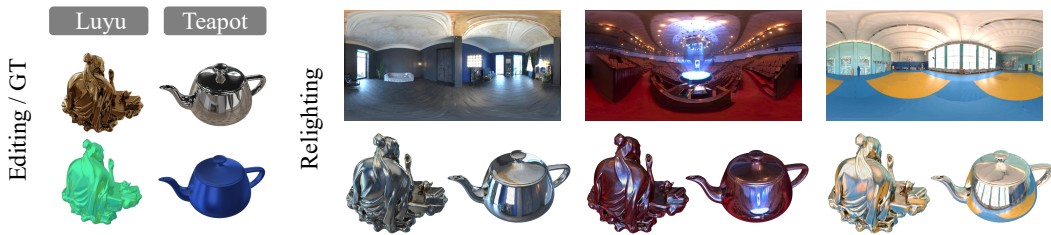

Figure 11: Relighting and editing on Luyu and Teapot with ground truth on the left.

| Ground Truth | Rendering | Normal | Metallic |
|:---:|:---:|:---:|:---:|
| **Specular lighting** | **Diffuse lighting** | **Indirect lighting** | **Visibility** |

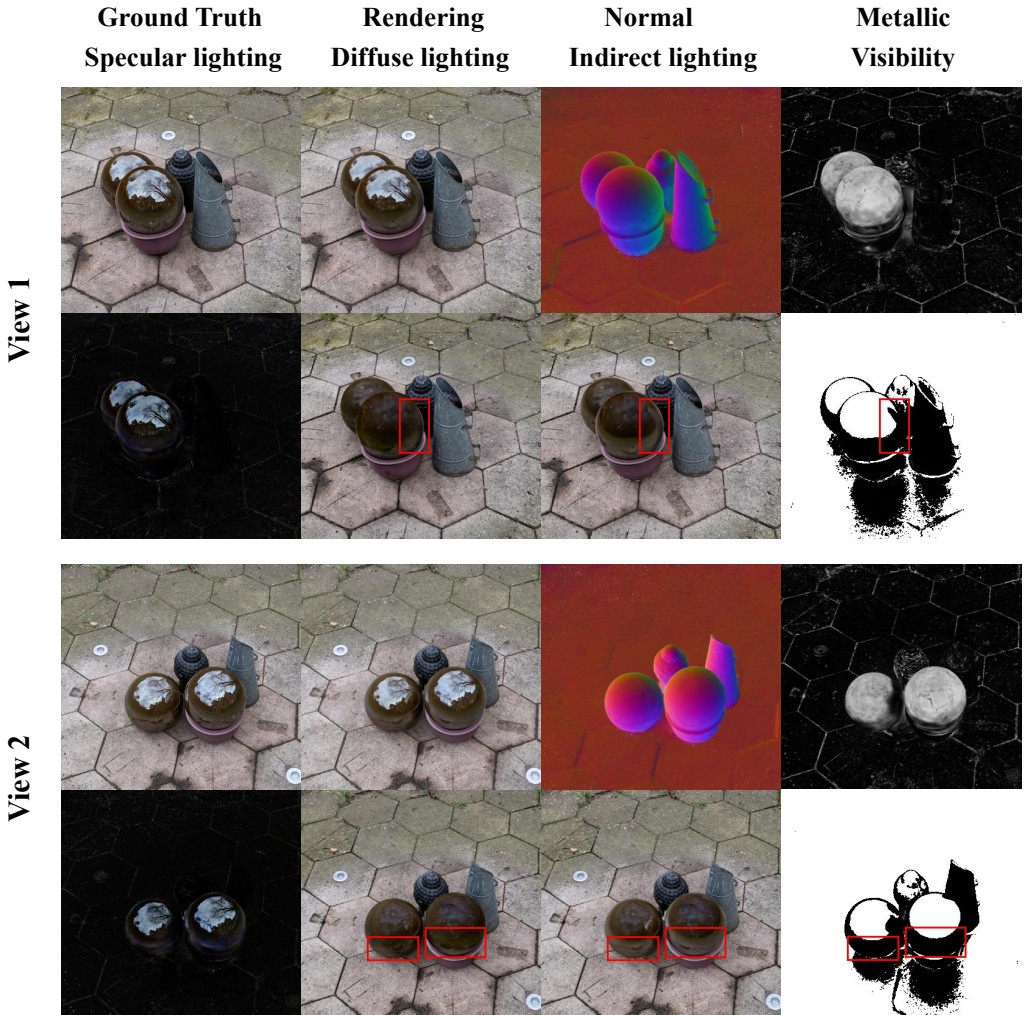

Figure 12: Qualitative results of indirect lighting on Ref-Real dataset (Verbin et al., 2022). **Indirect lighting**: Only considering indirect light as specular component when rendering - a focused view.

Table 8: Quantitative comparisons on NeRF-Synthetic dataset (Mildenhall et al., 2020).

| | chair | drums | ficus | hotdog | lego | material | ship |
|:---|:---:|:---:|:---:|:---:|:---:|:---:|:---:|
| **PSNR ↑** | | | | | | | |
| 3DGS | 35.03 | 26.04 | 35.29 | 37.57 | 33.71 | 30.04 | 31.43 |
| 3DGS-DR | 32.10 | 25.31 | 28.03 | 35.58 | 32.94 | 28.35 | 29.07 |
| *Ref-Gaussian* | 34.71 | 26.33 | 35.74 | 37.65 | 33.46 | 30.44 | 30.78 |
| **SSIM ↑** | | | | | | | |
| 3DGS | 0.987 | 0.954 | 0.987 | 0.985 | 0.975 | 0.959 | 0.906 |
| 3DGS-DR | 0.977 | 0.946 | 0.963 | 0.982 | 0.978 | 0.950 | 0.894 |
| *Ref-Gaussian* | 0.981 | 0.955 | 0.988 | 0.985 | 0.976 | 0.964 | 0.898 |
| **LPIPS ↓** | | | | | | | |
| 3DGS | 0.012 | 0.040 | 0.012 | 0.021 | 0.027 | 0.040 | 0.111 |
| 3DGS-DR | 0.024 | 0.055 | 0.055 | 0.033 | 0.026 | 0.044 | 0.129 |
| *Ref-Gaussian* | 0.021 | 0.039 | 0.012 | 0.025 | 0.026 | 0.037 | 0.125 |

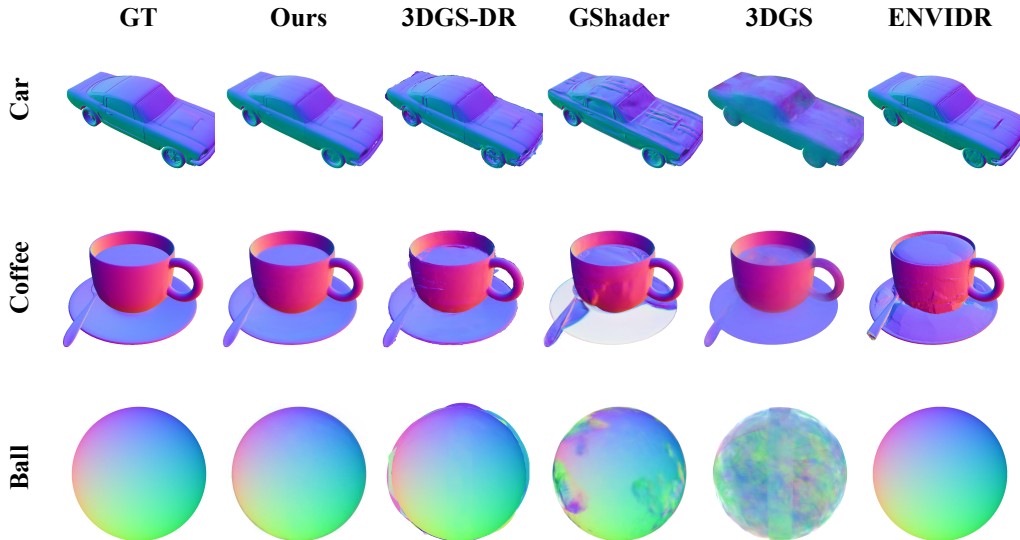

Figure 13: Per-scene qualitative comparisons of normals.

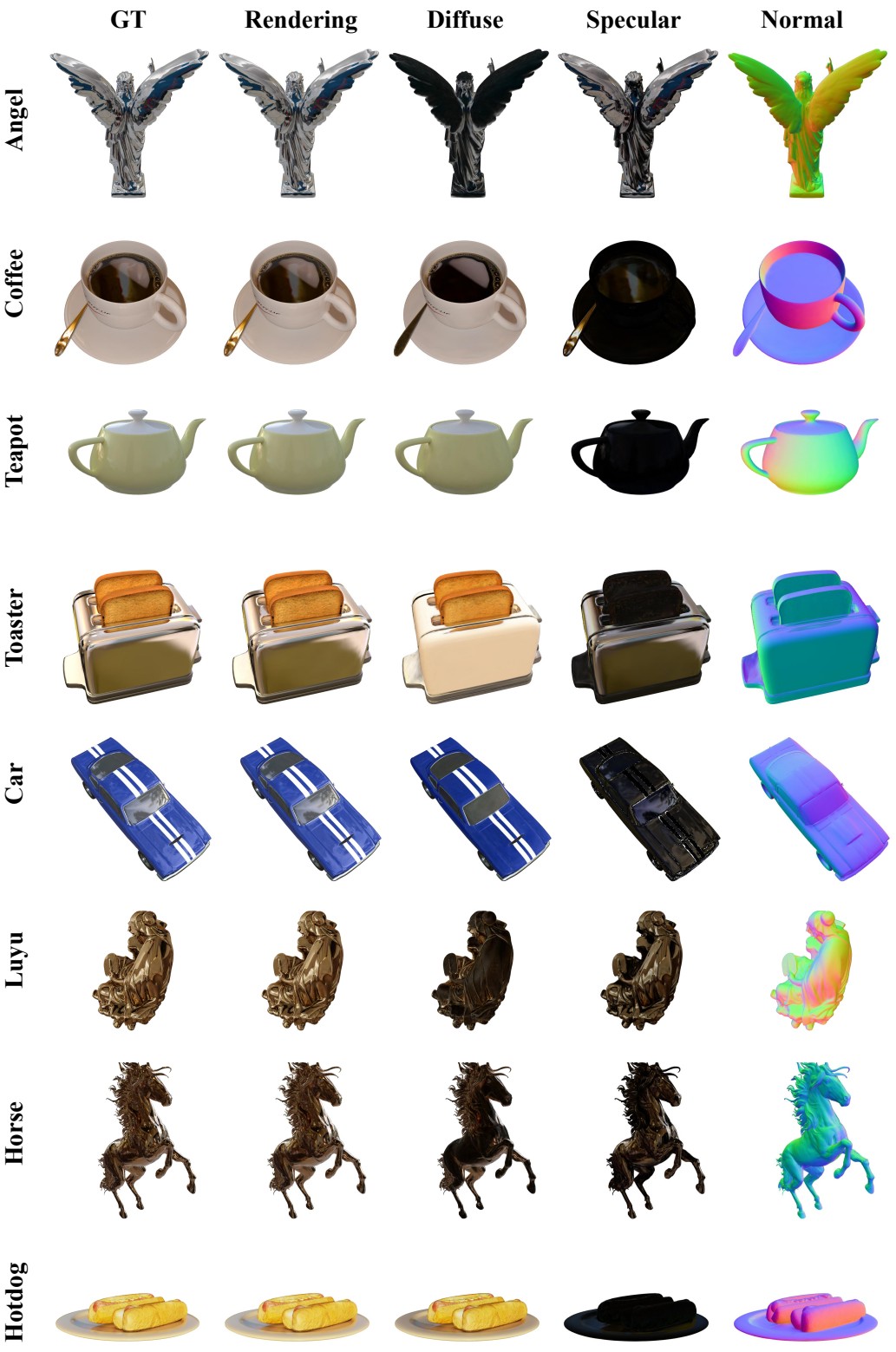

Figure 14: Qualitative decomposition results.