# OpenReview forum: "Reflective Gaussian Splatting"
_ICLR.cc/2025/Conference — ICLR 2025 Poster_

### Official Review · Reviewer_xuc5 · 2024-10-27

**Soundness:** 3
**Presentation:** 4
**Contribution:** 3
**Rating:** 6
**Confidence:** 4

**Summary:**

This paper focuses on novel view synthesis for reflective objects. It proposes a method called Reflective Gaussian Splatting, which uses Gaussian Splatting as its primary representation and introduces two additional techniques: (a) physically-based deferred shading and (b) an accurate approximation for modeling inter-reflection effects within the Gaussian Splatting framework. The effectiveness of this method is validated using various benchmark datasets. Additionally, the method is compute-efficient, making it suitable for applications such as relighting and material editing.

**Strengths:**

1. The proposed method achieves state-of-the-art performance on several benchmark datasets, validating its significance.
2. The proposed method offers a good trade-off between performance and efficiency. The introduced deferred shading technique, plus the occlusion approximation using a TSDF mesh, sounds reasonable for these goals.
3. The paper is well-written and easy to follow. The paper has enough details for reproduction.

**Weaknesses:**

1. While efficient in modeling the occlusion with an extracted mesh, its performance may rely on the accuracy of the extracted mesh. Since extracting a good mesh for the reflective object is non-trivial,  such a non-differentiable approximation may lead to worse results.

2. While built upon 3DGS-DR, its performance (LPIPS) on real datasets appears limited over 3DGS-DR.


#minors:
The citation in L49~L50 seems misplaced.

**Questions:**

1. From the paper, I assume that the geometry is important for disentangling appearance and those PBR materials. I would like to know if PBR modeling will also benefit geometry.

2. Why is the LIPPS metric of the proposed method on the real dataset worse than 3DGS-DR?

---

> ### Author Response · Authors · 2024-11-22
> **Reply to Reviewer xuc5**
>
> ## W1: Non-Differentiable Approximation of Mesh
> Thanks to the reviewer for the insightful comments. Extracting a mesh is non-trivial and crucial for visibility approximation, further highlighting the importance of accurate geometry reconstruction. To achieve this, we leverage a 2DGS ([1])framework (Figure 8), an initial stage with per-Gaussian shading, and material-aware normal propagation (Figure 10). Quantitative comparisons of surface normals (Table 3 in the revised manuscript, also shown below) further demonstrate their effectiveness. Since the non-differentiable approximation, TSDF, is independent and replaceable in our approach, we are pleased that we can find out more accurate and efficient techniques in our future work.
>
> ---
>
> ## Q1: Geometry Improvement Introduced by PBR
> We believe that our physically based rendering (PBR) method has contributed significantly to geometry improvement. Quantitative comparisons, specifically the MAE value of the model without PBR (w/o PBR in Table 3 of the revised manuscript), clearly demonstrate its notable geometric shortcomings compared to PBR.
>
> ---
>
> ## W2 & Q2: LPIPS Limited Over 3DGS-DR
> We appreciate the question being raised. The real-world dataset is highly complex but has relatively poor reflective details, which contrasts with the typical scenarios involving highly reflective surfaces that Ref-Gaussian focuses on. As such, this dataset is not ideal enough for testing reflectiveness. Furthermore, LPIPS is very sensitive to subtle color and texture mismatches that are insignificant for reconstruction, such as ground details. Other than LPIPS on the real-world dataset, Ref-Gaussian outperforms 3DGS-DR  ([2]) across all other metrics and datasets.
>
> ---
> ***Table 3: Comparison of MAE values across components of Ref-Gaussian.***
>
> | **Model**                   | **MAE (avg)** |
> |-----------------------------|---------------|
> | **Ref-Gaussian**                | 2.15 |
> | **w/o 2DGS**                | 4.45          |
> | **w/o Initial stage**       |  3.53 |
> | **w/o Material-aware**      | 3.86          |
> | **w/o PBR**                 | 3.81          |
>
>
>
> **Table 4: Ablation studies on the components of Ref-Gaussian.**
> *w/o Material-aware: use previous normal propagation following 3DGS-DR.*
>
> | **Model**                | **Ref-Gaussian** | **w/o PBR** | **w/o Inter-reflection** |  **w/o Deferred rendering** | **w/o Initial stage** | **w/o Material-aware** |
> |---------------------------|-------------|--------------------------|--------------|----------------------------|-----------------------|-------------------------|
> | **PSNR↑**               | 30.33 | 29.84       |  30.14 | 28.93       | 28.85                      | 29.94                | 29.68                  |
> | **SSIM↑**               | 0.958  | 0.956       |  0.957         | 0.935                      | 0.956                | 0.952                  |
> | **LPIPS↓**              | 0.049  | 0.052       |  0.050        | 0.074                      | 0.052                | 0.056                  |
>
>
> ### References
> 1. B. Huang, Z. Yu, A. Chen, A. Geiger, and S. Gao. *2D Gaussian Splatting for Geometrically Accurate Radiance Fields.* In SIGGRAPH, 2024.
> 2. K. Ye, Q. Hou, and K. Zhou. *3D Gaussian Splatting with Deferred Reflection.* In SIGGRAPH, 2024.

---

> > ### Comment · Reviewer_xuc5 · 2024-11-26
> > **Official Comment by Reviewer by xuc5**
> >
> > I think my concerns have been addressed. Even though some components introduced are not new, the current method would benefit the community. I keep my rating for acceptance and encourage the authors to open-source the code.

---

> > > ### Author Response · Authors · 2024-11-26
> > >
> > > We appreciate the reviewer's time for reviewing and thanks again for the valuable comments and the positive score!
> > > We will make our code and models publicly available to foster any further research.

---

> ### Author Response · Authors · 2024-11-25
>
> Dear Reviewer xuc5,
>
> We sincerely appreciate the reviewer's for reviewing, and we really want to have a further discussion with the reviewer to see if our response solves the concerns. We have addressed all the thoughtful questions raised by the reviewer *(eg,  non-differentiable approximation (TSDF), LPIPS and PBR's contribution to geometry)* and we hope that our work’s impact and results are better highlighted with our responses. It would be great if the reviewer can kindly check our responses and provide feedback with further questions/concerns (if any). We would be more than happy to address them. Thank you!
>
> Best wishes,
>
> Authors

---

### Official Review · Reviewer_9fZv · 2024-10-28

**Soundness:** 3
**Presentation:** 3
**Contribution:** 2
**Rating:** 6
**Confidence:** 5

**Summary:**

This paper proposes a Reflective Gaussian splatting (*Ref-Gaussian*) framework to achieve real-time high-level novel view synthesis for reflective objects with inter-reflection effects. The framework consists of 2 main parts: (i) physically-based deferred rendering, which associates each Gaussian with material properties and leverages physically-based surface rendering (rendering equation and split-sum approximation) to compute the rendered color; (ii) Gaussian-grounded inter-reflection, which computes the visibility of the reflected ray and model indirect view-dependent color as a per-Gaussian spherical harmonics. The proposed method also leverages several techniques to enhance the underlying geometry. This paper evaluates the proposed method with qualitative and quantitative experimental results and validates its design components by an ablation study. It shows that the method can outperform baselines.

**Strengths:**

1. The key designs proposed by the paper are rational. It is not surprising that leveraging physically-based rendering can significanty improve the performance of specular rendering.
2. I like the idea of material-aware normal propagation. Seems like it can greatly improve the quality of surface normal reconstruction.
3. Extensive qualitative and quantitative experiments demonstrate that the proposed method outperforms baselines.
4. I appreciate the comprehensive ablation study that covers lots of specific design choices.

**Weaknesses:**

1. Missing details. L233-234: "During optimization, we periodically extract the object’s surface mesh using truncated signed distance function (TSDF) fusion." The authors need to specify the number of steps as the period of mesh extraction.
2. Unclear explanations. I have several confusions when reading the paper, and please see the "Questions" part. I believe that the paper writing can be improved to explain the method more clearly.
3. Baselines. Although this method is based on 3DGS, it should also include more methods based on other representations (such as NeRF) as its baseline, such as (a) NeRO: Neural Geometry and BRDF Reconstruction of Reflective Objects from Multiview Images (SIGGRAPH 2023) (b) Neural Directional Encoding for Efficient and Accurate View-Dependent Appearance Modeling (CVPR 2024).
4. Limitation of Gaussian-grounded inter-reflection: In Eq. (8), it is clear that the direct lighting part can capture rough specular effect. But in Eq. (10), the method only traces a single reflected ray to compute the indirect reflection, which will introduce errors for rough surfaces. This should be added as a limitation.
5. Experiments. The method leverages the extracted mesh to compute visibility, so I think the author should also show (at least qualitative) results of the extracted mesh in the experiment section.

I may consider increasing the rating if my concerns and questions are addressed in the rebuttal.

**Questions:**

1. L239-244. More details are required to describe how the indirect lighting is rendered. To be specific,
    - Does the method conducts explicit ray tracing for the reflected ray and computes ray-Gaussian intersection? Or does the method uses Gaussian splatting in the reflected ray directions?
    - In Eq. (10), the meaning of the symbol $N$ in $i\in N$ is unclear: does it denote the set of Gaussians intersected with the reflected ray?
2. L254: "To mitigate this, we apply the rendering equation at the Gaussian level to achieve geometry convergence initially." What does "rendering equation at the Gaussian level" mean? What is the difference between this and Eq. (8)?
3. Are the training hyperparameters the same across different scenes?

---

> ### Author Response · Authors · 2024-11-22
> **Reply to Reviewer 9fZv**
>
> ## W1: Mesh Extracted Steps Interval
> During optimization, we extract the object’s surface mesh every 3000 steps using truncated signed distance function (TSDF) fusion. We have updated this in the revised manuscript.
>
> ---
>
> ## W2: Unclear Explanations
> We apologize for the confusion. Our further explanations are as follows:
>
> ### Q1: Inter-reflection Rendering Details
> 1. We perform ray tracing on the extracted mesh for the reflected ray **R**:
> $$
>    \mathbf{R} = 2(w_i \cdot N)N - w_i
> $$
>    to determine whether it is occluded, where $w_i$ represents the light direction and $N$ represents the normal. The mesh is extracted using truncated signed distance function (TSDF) fusion. Ray tracing is conducted for ray-triangle intersections rather than ray-Gaussian intersections, as the latter can be significantly slower due to the need for alpha-blending to accumulate opacity.
>
> 2. Regarding N in Eq. 10, the formula has been clarified and updated in the revised manuscript as:
> $$
>    L_\mathrm{ind} = \sum_{i=1}^{N} l_\mathrm{ind} \alpha_i\prod_{j=1}^{i-1} (1 - \alpha_j),
> $$
>
> ### Q2: Initial Stage with Per-Gaussian Shading
> In a typical rendering process, the integral in Eq. 9 is computed using aggregated pixel-level feature maps (e.g., albedo, metallic, roughness, and normal). However, this approach can hinder geometry convergence in the early stages of optimization due to less effective gradients introduced by deferred shading. To address this, we propose an initial stage where the rendering equation is applied directly to each Gaussian, using the material and geometry properties associated with it to compute the outgoing radiance. The outgoing radiance is then alpha-blended during rasterization to produce the final physically based rendering (PBR). This adjustment significantly accelerates geometry convergence and improves overall quality, as shown in the ablation studies (Table 3 and Table 4, in the revised manuscript, also shown below).
>
> ### Q3: Hyperparameters
> Our hyperparameters remain consistent across all datasets.
>
> ---
>
> ## W3: Missing Baselines of More NeRF-Based Methods
> Thank you for this valuable suggestion. In addition to 3DGS-based competitors, we have included two NeRF-based competitors, RefNeRF ([1]) and ENVIDR ([2]), in Table 1. To address your concern, we have added comparisons in Table 5 and Table 6 (Appendix of the revised manuscript). These include metrics such as PSNR, SSIM, LPIPS, and FPS on the Glossy Blender dataset ([3]) with NeRO ([3]) and NeuS ([4]), as well as with NDE ([5]) on the Shiny Blender dataset ([1]). The results demonstrate that Ref-Gaussian significantly outperforms NeRO and remains comparable to NDE in rendering quality while achieving much better rendering speed and training efficiency.
>
> ---
>
> ## W4: Limitation of Gaussian-Grounded Inter-reflection
> Thank you for your insightful comments. Ref-Gaussian primarily focuses on reconstructing highly reflective objects. For efficiency, we compute indirect lighting using a single reflected ray. While it is possible to trace additional rays using Monte Carlo sampling for more accurate representation of indirect lighting, this approach would significantly reduce efficiency. This limitation has been discussed in the revised manuscript.
>
> ---
>
> ## W5: Further Qualitative Results of the Extracted Mesh
> Following your suggestion, we have included a visualization of the extracted mesh in Figure 5 of the revised manuscript.

---

> ### Author Response · Authors · 2024-11-22
> **Reply to Reviewer 9fZv**
>
> ***Table 3: Comparison of MAE values across components of Ref-Gaussian.***
>
> | **Model**                   | **MAE (avg)** |
> |-----------------------------|---------------|
> | **Ref-Gaussian**                | 2.15 |
> | **w/o 2DGS**                | 4.45          |
> | **w/o Initial stage**       |  3.53 |
> | **w/o Material-aware**      | 3.86          |
> | **w/o PBR**                 | 3.81          |
>
>
>
> **Table 4: Ablation studies on the components of Ref-Gaussian.**
> *w/o Material-aware: use previous normal propagation following 3DGS-DR.*
>
> | **Model**                | **Ref-Gaussian** | **w/o PBR** | **w/o Inter-reflection** |  **w/o Deferred rendering** | **w/o Initial stage** | **w/o Material-aware** |
> |---------------------------|-------------|--------------------------|--------------|----------------------------|-----------------------|-------------------------|
> | **PSNR↑**               | 30.33 | 29.84       |  30.14 | 28.93       | 28.85                      | 29.94                | 29.68                  |
> | **SSIM↑**               | 0.958  | 0.956       |  0.957         | 0.935                      | 0.956                | 0.952                  |
> | **LPIPS↓**              | 0.049  | 0.052       |  0.050        | 0.074                      | 0.052                | 0.056                  |
>
>
> ### References
> 1. D. Verbin, P. Hedman, B. Mildenhall, T. Zickler, J. T. Barron, and P. P. Srinivasan. *Ref-NeRF: Structured View-Dependent Appearance for Neural Radiance Fields.* In CVPR, 2022.
> 2. R. Liang, H. Chen, C. Li, F. Chen, S. Panneer, and N. Vijaykumar. *ENVIDR: Implicit Differentiable Renderer with Neural Environment Lighting.* In ICCV, 2023.
> 3. Y. Liu, P. Wang, C. Lin, X. Long, J. Wang, L. Liu, T. Komura, and W. Wang. *NeRO: Neural Geometry and BRDF Reconstruction of Reflective Objects from Multi-View Images.* ACM Trans. Graph., 2023.
> 4. P. Wang, L. Liu, Y. Liu, C. Theobalt, T. Komura, and W. Wang. *NeuS: Learning Neural Implicit Surfaces by Volume Rendering for Multi-View Reconstruction.* arXiv preprint, 2021.
> 5. L. Wu, S. Bi, Z. Xu, F. Luan, K. Zhang, I. Georgiev, K. Sunkavalli, and R. Ramamoorthi. *Neural Directional Encoding for Efficient and Accurate View-Dependent Appearance Modeling.* In CVPR, 2024.

---

> ### Comment · Reviewer_9fZv · 2024-11-24
>
> Thank you for your reply! Most of my questions have been addressed, but I still have 2 remaining questions below:
> 1. **Limitation of Gaussian-Grounded Inter-reflection**: As you acknowledged, Ref-Gaussian primarily focuses on reconstructing highly reflective objects due to its single reflected ray tracing design. This is one of my major concerns about the limitations of this paper, as most common objects are not purely highly reflective, which restricts the application scenario of this method.
> 2. **Request for more qualitative results**: The results (especially geometry results) shown in this paper are impressive, and I'd like to see qualitative results of more scenes included in the appendix. It would be nice if the authors could provide per-scene qualitative results. If the per-scene results are satisfactory, I may consider raising my rating.

---

> ### Author Response · Authors · 2024-11-25
> **Reply to Reviewer 9fZv**
>
> Thanks again for reviewer's valuable time and efforts to review our work as well as the insightful comment.
>
> **1. Limitation of Gaussian-grounded inter-reflection**
>
> Thanks for raising this issue which we should have elaborated and clarified in more details. As mentioned in Section 3.2 of the main paper, Ref-Gaussian calculates indirect lighting using a single reflected ray, making it particularly effective and efficient in reflective scenes. Also, as described in Eq.9, inter-reflection is incorporated as additional element into the specular component. Consequently, in less reflective scenes, the influence of inter-reflection diminishes naturally and accordingly as the metallic attribute decreases. In these cases, the diffuse component takes over and captures most of the inter-reflection effects. This behavior is evident in the decomposition results provided in Figure 5 and Figure 14 (appendix). As a result, **Ref-Gaussian maintains robust rendering quality for less reflective scenes, providing a unified solution for both reflective and non-reflective scenes**. In contrast, previous reflection focused alternatives such as 3DGS-DR ([1]) are designed specifically only for reflective scenes, leading to narrowed applications as mentioned. To support this claim, we have included Table 8 in the appendix, which presents a per-scene quantitative comparison among Ref-Gaussian, 3DGS-DR ([1]), and 3DGS ([2]) on the nerf-synthetic dataset, predominantly composed of non-reflective scenes, for novel view synthesis. The results show that our Ref-Gaussian still excels over both alternatives, validating its generic and unified advantages.
>
>
>
> **2. Request for more qualitative results**
>
> Following the reviewer's suggestion, we provide further illustration of our advantage on geometry reconstruction in Figure 13 and 14 of the appendix. The qualitative comparison in Figure 13 demonstrates Ref-Gaussian's comprehensive grasp of details over the alternatives (such as the tires in car and the water surface in coffee).
> To be clear, please refer to our video demo in the supplementary material for best view of our qualitative results.
>
>
> Hope our responses clarify above thoughtful questions, and it is very much appreciated if the reviewer can kindly check our responses and provide feedback with further questions/concerns (if any). We would be more than happy to address them. Thank you!
>
>
> **References**
> > 1. K. Ye, Q. Hou, and K. Zhou. 3d gaussian splatting with deferred reflection. In SIGGRAPH, 2024.
> > 2. B. Kerbl, G. Kopanas, T. Leimkühler, and G. Drettakis. *3D Gaussian Splatting for Real-Time Radiance Field Rendering.* ACM Trans. Graph., 2023.

---

> > ### Comment · Reviewer_9fZv · 2024-11-25
> >
> > Thank you for your reply. I think the additional results are satisfactory for me, and I will increase my rating toward marginal accept.

---

> > > ### Author Response · Authors · 2024-11-25
> > >
> > > We appreciate the reviewer's time for reviewing and thanks again for the valuable comments and the positive score!

---

### Official Review · Reviewer_voQB · 2024-11-02

**Soundness:** 2
**Presentation:** 3
**Contribution:** 2
**Rating:** 6
**Confidence:** 4

**Summary:**

This paper proposed a novel Gaussian Splatting based inverse rendering framework, which focus on reflective object reconstruction. This method rely on deferred shading to achieve more smooth and cohesive rendering results, as well as the combination of mesh-based visibility and per-Gaussian indirect lighting to model the inter-reflection. The experimenta evaluation demonstrate that this method can accurately reconstruct reflective object while maintaining real-time rendering capabilities.

**Strengths:**

- Rendering quality is excellent, and reconstructed normals are accurate.
- Training time and rendering speed are satisfactory

**Weaknesses:**

- The contribution of this paper may lack novelty. For the deferred shading part of the pipeline, I think many previous 3DGS-based inverse rendering methods[1][2][3] have adopted these techniques and [2][3] also use split-sum approximation to handle the intractable rendering equation. Further more, assigning each Gaussian with a new attribute to model the indirect lighting has also been proposed in previous methods[4]. The innovation of this method lies in the new visibility modeling scheme and optimization techniques. Unlike previous methods that use baked volume or Gaussian-based ray-tracing to model occlusion, the proposed method attempts to first extract the mesh using TSDF and then use mesh-based ray-tracing to obtain occlusion.

- To determine the visibility, this method consider the occlusion at the reflected direction $\boldsymbol{R}=2\left(w_i \cdot \boldsymbol{N}\right) \boldsymbol{N}-w_i$. This means that this method only considers the indirect lighting of the specular surface. For glossy or diffuse surfaces, this estimation may not be not accurate enough, and such objects do exist in the dataset. For example, for the Potion in figure.5, The lid of the bottle is obviously a rough diffuse surface. Since the proposed method focuses on the reconstruction of the reflective object, this does not seem to be a serious problem, but I would like to know if there is a way to improve this.

- In addition, since the visibility only considers the reflected direction, I am a little confused about the integral in equation 9. Because this method does not calculate the integral over the entire hemisphere $\Omega$. So I want to know how the rendering equation is finally calculated.





[1] DeferredGS: Decoupled and Editable Gaussian Splatting with Deferred Shading https://arxiv.org/abs/2404.09412

[2] GS-IR: 3D Gaussian Splatting for Inverse Rendering https://openaccess.thecvf.com/content/CVPR2024/papers/Liang_GS-IR_3D_Gaussian_Splatting_for_Inverse_Rendering_CVPR_2024_paper.pdf

[3] 3DGaussian Splatting with Deferred Reflection https://dl.acm.org/doi/pdf/10.1145/3641519.3657456

[4] Relightable 3D Gaussians: Realistic Point Cloud
 Relighting with BRDF Decomposition and Ray
 Tracing https://www.ecva.net/papers/eccv_2024/papers_ECCV/papers/06121.pdf

**Questions:**

- In ablation study, the normal of 2DGS is much better than 3DGS. Since accurate normals are very important for PBR, does this mean that the performance gain of the proposed method over other methods comes largely from the better geometry reconstruction quality of 2DGS? This is important for evaluating the technical contribution of this paper. I hope the author can give quantitative data (**use 3DGS as representation and keep the rest of the pipeline unchanged to evaluate the rendered results**) to show how much performance improvement can be achieved by using 2DGS representation compared to 3DGS.
- Since indirect lighting is modeled as an attribute of each Gaussian, which represents the inter-reflection under the lighting conditions corresponding to the training stage. Is it possible to build indirect lighting when relighting?

---

> ### Author Response · Authors · 2024-11-22
> **Reply to Reviewer voQB**
>
> ## W1: Contributions
> We thank the reviewer for raising this question! While 3DGS-DR ([1]) utilizes deferred rendering, it employs a simple shading model that does not consider physically based rendering, resulting in inaccurate estimations. Additionally, it is unable to model indirect lighting. GS-IR ([2]), which also uses split-sum approximation to address the rendering equation, relies on spherical harmonics to represent averaged occlusion and indirect lighting for the diffuse component. However, this approach leads to inaccurate diffuse estimations and does not account for indirect lighting in the specular term, making it unsuitable for accurately modeling highly reflective objects. DeferredGS ([3]) also uses split-sum approximation but lacks the ability to model indirect lighting. Furthermore, it requires joint training with an "Instant-RefNeuS" model, complicating the optimization process. On the other hand, R3DG ([4]) models indirect lighting using Monte Carlo sampling for the rendering equation. While this improves accuracy, it significantly slows down rendering speed. Moreover, its visibility computation relies on ray tracing within the 3DGS ([5]) framework, which is inherently imprecise due to 3DGS's lack of a clear definition in 3D space, as 3DGS is trained in 2D space.
>
> To address these limitations, our method proposes a unified approach that integrates deferred rendering, physically based rendering, and ray tracing on meshes for visibility, along with simultaneous modeling of indirect lighting.  Additionally, we introduce several techniques to enhance geometry reconstruction, including a 2DGS ([6]) framework, an initial stage with per-Gaussian shading, and material-aware normal propagation. Other than a simple combination, each one of the techniques as mentioned above is properly innovated and adjusted to the best of Ref-Gaussian, enabling our method to handle highly reflective objects more accurately and more effectively.
>
> ---
>
> ## W2: Indirect Lighting for the Diffuse Term
> Thank you for this insightful comment. Since the diffuse component is not sensitive to viewing direction, we use spherical harmonics to directly model its outgoing radiance, as discussed in the implementation details. This approach enables the model to account for both occlusion and indirect lighting effects. For the specular component, our method focuses on highly reflective objects and computes visibility by tracing a single ray along the reflected direction. For glossy objects, Monte Carlo sampling within the specular lobe could be employed to trace multiple rays for improved visibility estimation. However, this approach would come at the cost of reduced rendering speed.
>
> ---
>
> ## W3: Integral of the Specular Term
> We apologize for the confusion. As explained in our paper, to avoid the high computational cost of Monte Carlo sampling, we adopt the split-sum approximation as described in Eq. 8. In this formulation, the left term depends solely on $(\omega_i \cdot N)$ and roughness $R$, while the right term, denoted as $L_\mathrm{dir}$, represents the integral of radiance over the specular lobe. To incorporate indirect lighting, we modify the right term $L_\mathrm{dir}$ in Eq. 8 to $L_\mathrm{dir}V + L_\mathrm{ind}(1-V)$, where $L_\mathrm{ind}$ models the indirect lighting for the reflected direction only. Here, $V$ is the visibility term computed via ray tracing on the extracted mesh in the reflected direction, and $L_\mathrm{ind}$ is calculated using Eq. 10. This approach provides an efficient balance between computational cost and the modeling of indirect lighting.
>
> ---
>
> ## Q1: Geometry Improvements Using 2DGS Representation
> Thank you for raising this important question! Accurate geometric reconstruction is crucial for the effective performance of the proposed physically based deferred rendering and inter-reflection methods. To this end, we implemented a series of geometric optimizations, including replacing 3DGS with 2DGS. Our existing quantitative(Table 3,4) and qualitative(Figure 7,9) ablation studies in our revised manuscript demonstrate that both the physically based deferred rendering and inter-reflection methods significantly enhance rendering quality. Following your suggestion, we have added quantitative ablation specifically related to the impact of 2DGS(Table 7, Appendix of our revised manuscript). In this additional analysis, we used 3DGS as the representation while keeping the rest of the pipeline unchanged to evaluate the rendered results. Metric comparisons for both geometric reconstruction and novel view synthesis performance have been included in the revised manuscript, as shown in Table 3 and Table 4 which are also shown below.

---

> ### Author Response · Authors · 2024-11-22
> **Reply to Reviewer voQB**
>
> ## Q2: Building Indirect Lighting When Relighting
> Thank you for your valuable suggestions. Recent progress based on 3DGS has not yet achieved the ability to build indirect lighting when relighting. Relighting requires querying arbitrary light in the space, which is difficult to achieve using the splatting technique due to its limitations inherited from rasterization. However, the Gaussian ray tracing method proposed by 3DGRT ([7]) suggests an alternative to rasterization for solving this problem. This marks a breakthrough in rendering ray-based effects and provides clear guidance for our future work.
>
> ---
>
> ***Table 3: Comparison of MAE values across components of Ref-Gaussian.***
>
> | **Model**                   | **MAE (avg)** |
> |-----------------------------|---------------|
> | **Ref-Gaussian**                | 2.15 |
> | **w/o 2DGS**                | 4.45          |
> | **w/o Initial stage**       |  3.53 |
> | **w/o Material-aware**      | 3.86          |
> | **w/o PBR**                 | 3.81          |
>
>
>
> **Table 4: Ablation studies on the components of Ref-Gaussian.**
> *w/o Material-aware: use previous normal propagation following 3DGS-DR.*
>
> | **Model**                | **Ref-Gaussian** | **w/o PBR** | **w/o Inter-reflection** |  **w/o Deferred rendering** | **w/o Initial stage** | **w/o Material-aware** |
> |---------------------------|-------------|--------------------------|--------------|----------------------------|-----------------------|-------------------------|
> | **PSNR↑**               | 30.33 | 29.84       |  30.14 | 28.93       | 28.85                      | 29.94                | 29.68                  |
> | **SSIM↑**               | 0.958  | 0.956       |  0.957         | 0.935                      | 0.956                | 0.952                  |
> | **LPIPS↓**              | 0.049  | 0.052       |  0.050        | 0.074                      | 0.052                | 0.056                  |
>
>
>
> **Table 7: Per-scene PSNR comparison on synthesized test views**
> *w/o 2DGS: Use 3DGS as representation and keep the rest of the pipeline unchanged.*
> | **Datasets** | Shiny Blender |       |        |        |        |         | Glossy Synthetic |       |       |       |       |        |       |        |
> | ------------ | ------------- | ----- | ------ | ------ | ------ | ------- | ---------------- | ----- | ----- | ----- | ----- | ------ | ----- | ------ |
> | **Scenes**   | ball          | car   | coffee | helmet | teapot | toaster | angel            | bell  | cat   | horse | luyu  | potion | tbell | teapot |
> | ENVIDR       | 41.02         | 27.81 | 30.57  | 32.71  | 42.62  | 26.03   | 29.02            | 30.88 | 31.04 | 25.99 | 28.03 | 32.11  | 28.64 | 26.77  |
> | 3DGS-DR      | 33.43         | 30.48 | 34.53  | 31.44  | 47.04  | 26.76   | 29.07            | 30.60 | 32.59 | 26.17 | 28.96 | 32.65  | 29.03 | 25.77  |
> | w/o 2DGS     | 36.10         | 30.65 | 34.51  | 33.29  | 44.25  | 27.03   | 28.33            | 30.60 | 33.14 | 26.70 | 29.35 | 32.94  | 29.17 | 26.31  |
> | **Ref-Gaussian**    | 37.01         | 31.04 | 34.63  | 32.32  | 47.16  | 28.05   | 30.38            | 32.86 | 33.01 | 27.05 | 30.04 | 33.07  | 29.84 | 26.68  |
>
>
>
>
>
> ### References
> 1. K. Ye, Q. Hou, and K. Zhou. *3D Gaussian Splatting with Deferred Reflection.* In SIGGRAPH, 2024.
> 2. Z. Liang, Q. Zhang, Y. Feng, Y. Shan, and K. Jia. *GS-IR: 3D Gaussian Splatting for Inverse Rendering.* In CVPR, 2024.
> 3. T. Wu, J.-M. Sun, Y.-K. Lai, Y. Ma, L. Kobbelt, and L. Gao. *DeferredGS: Decoupled and Editable Gaussian Splatting with Deferred Shading,* 2024b.
> 4. J. Gao, C. Gu, Y. Lin, H. Zhu, X. Cao, L. Zhang, and Y. Yao. *Relightable 3D Gaussian: Real-Time Point Cloud Relighting with BRDF Decomposition and Ray Tracing.* arXiv preprint, 2023.
> 5. B. Kerbl, G. Kopanas, T. Leimkühler, and G. Drettakis. *3D Gaussian Splatting for Real-Time Radiance Field Rendering.* ACM Trans. Graph., 2023.
> 6. B. Huang, Z. Yu, A. Chen, A. Geiger, and S. Gao. *2D Gaussian Splatting for Geometrically Accurate Radiance Fields.* In SIGGRAPH, 2024.
> 7. N. Moënne-Loccoz, A. Mirzaei, O. Perel, R. de Lutio, J. M. Esturo, G. State, S. Fidler, N. Sharp, and Z. Gojcic. *3D Gaussian Ray Tracing: Fast Tracing of Particle Scenes.* arXiv preprint, 2024.

---

> > ### Comment · Reviewer_voQB · 2024-11-25
> > **Official Comment by Reviewer voQB**
> >
> > I appreciate the authors for providing such a detailed explanation in response to my questions. And the authors' response have already address most of my concerns. Besides, additional ablation study shows that even without using 2DGS, a stronger representation in geometric reconstruction, the proposed method can still achieve good performance, which proves the effectiveness of the training strategy proposed in the paper. Therefore, I will raise my rating. But I still have some questions about Eq (8). Why do we need to multiply $L_{ind}$ by $(1-V)$? In my understanding, $L_{ind}$ is the sum of incident radiance in all occlusion directions, so there is no need to consider visibility. But as far as I know, the same approach is also used in GS-IR, so I would like the author to explain this point.

---

> > > ### Author Response · Authors · 2024-11-26
> > > **Reply to voQB**
> > >
> > > Thanks again for the reviewer's valuable time and efforts for reviewing our work, as well as the insightful comments. We want to express our sincere gratitude for the recognition of our work.
> > >
> > > Regarding the indirect component $L_\mathrm{ind}$, to further clearity, we want to first note that $L_\mathrm{ind}$ is an newly introduced view-dependent color component represented using spherical harmonics and $V$ is a binary mask, controling the selection between $L_\mathrm{ind}$ and $L_\mathrm{dir}$. We don't quite understand the comment of "$L_\mathrm{ind}$ is the sum of incident radiance in all occlusion directions", and please help kindly further explain if still necessary and we will follow up.
> > >
> > >
> > > On the necessity of the $1-V$ term for $L_\mathrm{ind}$, and the suggested form:
> > > $$
> > > L_s'(\omega_o) \approx ( \int_{\Omega} f_s(\omega_i, \omega_o) (\omega_i \cdot \mathbf{N}) d\omega_i ) \cdot [L_\mathrm{dir} \cdot V + L_\mathrm{ind}].
> > > $$
> > > This is equivalent to ours: When the visibility is set to 1, the model will optimize the value of indirect light to 0, which is identical to the Eq.9 of our paper. Thus, either form is adoptable.
> > >
> > > Lastly on the comparsion with GS-IR ([1]) which also uses spherical harmonics to represent indirect lighting for the diffuse component, we highlight these most related differences: (1) By using spherical harmonics to estimate a binary occlusion (i.e., the $1-V$ term in our method), additional errors may be introduced with GS-IR. Instead, we resort to the ray tracing which is more physically plausible. (2) Modeling indirect lighting for the diffuse component is ineffective for tackling reflection scenes as we focus on here, as discussed earlier, because, its effect is marginal.
> > >
> > >
> > > Hope our responses clarify the thoughtful questions above, and it would be very much appreciated if the reviewer could kindly check our responses and provide feedback with further questions or concerns (if any). We would be more than happy to address them. Thank you!
> > >
> > > **Reference**:
> > >
> > > [1] Z. Liang, Q. Zhang, Y. Feng, Y. Shan, and K. Jia. *GS-IR: 3D Gaussian Splatting for Inverse Rendering.* In CVPR, 2024.

---

> ### Author Response · Authors · 2024-11-25
>
> Dear Reviewer voQB,
>
> We sincerely appreciate the reviewer's time for reviewing, and we really want to have a further discussion with the reviewer to see if our response solves the concerns. We have addressed all the thoughtful questions raised by the reviewer *(eg, contributions, indirect lighting for the diffuse term, integral of the specular term, geometry improvements using 2DGS, building indirect lighting when relighting)* and we hope that our work’s impact and results are better highlighted with our responses. It would be great if the reviewer can kindly check our responses and provide feedback with further questions/concerns (if any). We would be more than happy to address them. Thank you!
>
> Best wishes,
>
> Authors

---

> ### Comment · Reviewer_voQB · 2024-11-26
>
> I still have some questions about Eq (9). According to Section 3.2, to model the inter-reflection, Reflective Gaussian Splatting estimates the binary visibility along the reflected direction $\boldsymbol{R}=2\left(w_i \cdot \boldsymbol{N}\right) \boldsymbol{N}-w_i$. In Eq (9), BRDF is integrated first and then multiplied by the incident light. But incident light comes from different directions, so how is V obtained?
>
> Besides, why consider visibilty along the reflected direction $R$ rather than the incident direction $\omega_i$? This does not conform to the definition of visibility in computer graphics. All previous inverse rendering methods, such as NeRV, InvRender, TensoIR, R3DG and GS-IR represents the visibility as $V\left(x,{\omega}_i\right)$. I think the author's definition of visibility is wrong, because for the incident light $L_i$, being occluded in the $R$ direction does not mean that the contribution in the direction $\omega_o$ is 0.
>
> Furthermore, I have a more important concern that the paper does not provide any quantitative comparison of relighting and BRDF estimation, which is crucial for evaluating the quality of inverse rendering work.

---

> ### Author Response · Authors · 2024-11-27
> **Reply to voQB**
>
> Thanks to the reviewer for the detailed review and the valuable insights the reviewer have provided. We sincerely apologize for the oversight in our manuscript, and we appreciate the careful attention to this detail.
>
> First and foremost, we made a mistake/typo in expressing the reflection direction and sincerely apologize for this. It should be $R=2(w_o\cdot {N}){N}-w_o$, rather than ${R}=2(w_i\cdot {N}){N}-w_i$ which caused the confusion of the reviewer. The revised formula is now fully aligned with what the reviewer commented. We are grateful for the reviewer's correction. This error has been rectified in our revised submission. But we apologize for forgetting to mark the corrections of L229-230 in blue.
>
> Please note, the primary goal of this work is novel view synthesis with reflective objects in the scene, rather than inverse rendering. And relighting is presented as an additional downstream task to demonstrate the versatility of our method, a unique feature as compared to previous reflection focused alternatives. While BRDF is part of our model, estimation of BRDF properties just represents the intermediate features (as shown in Figure 5) which facilitate the final novel view synthesis.
>
> We genuinely appreciate the reviewer's insightful comments and assistance in improving our work. Thanks once again for all the valuable feedback and understanding.

---

### Official Review · Reviewer_147t · 2024-11-03

**Soundness:** 3
**Presentation:** 4
**Contribution:** 3
**Rating:** 8
**Confidence:** 4

**Summary:**

Though powerful in novel view synthesis, vanilla 3DGS encounters challenges in extending to physically-based rendering or modeling inter-reflection due to lack of deterministic geometry. This work introduces a novel approach Ref-Gaussian, which achieves real-time high-quality rendering of reflective objects while also modeling inter-reflection effects.

The paper proposes several key techniques:
(a) Geometry enhanced technique: employing 2DGS to bridge deterministic geometry with Gaussian splatting and enhancing the geometry by the novel material-aware normal propagation.
(b) PBR optimization framework for 3DGS-based methods: using per-Gaussian PBR initialization followed with physically-based deferred rendering.
(c) Gaussian-grounded inter-reflection: applying real-time ray-tracing to the extracted mesh from 2DGS.

Extensive experiments demonstrate the effectiveness of these techniques and that the proposed method outperforms several baselines significantly.

**Strengths:**

1. By extracting explicit geometry, the paper addresses the inter-reflection issue in Gaussian splatting, which is important in realistic PBR. Experimental results showcase its SOTA performance in novel view synthesis and decomposition on reflective cases.
2. Instead of per-Gaussian PBR (like Relightable 3DGS or Gaussian Shader), The proposed method employs an effective PBR deferred rendering to achieve better PBR performance (similar to 3DGS-DR). Further ablation study demonstrates the superiority of such a deferred rendering technique over the per-Gaussian solutions.
3. The proposed method employs a material-aware normal propagation. This enhances normal estimation by periodically increasing the scale of 2D Gaussians with high metallic and low roughness, which demonstrates interesting material-normal interaction in 3DGS-based PBR.

**Weaknesses:**

1. The effectiveness of inter-reflection technique lacks further qualitative evidence (e.g. providing indirect components in Fig.5 or showcasing indirect components in Ref-Real dataset where the multiple objects provide rich inter-reflection), as the ablation study in table 3 indicates only a slight decrease in PSNR when rendering without inter-reflection.
2. The ablation study in table 3 only takes PSNR changes into account, while the influences on geometry may need further demonstration (e.g. normal MAE or qualitative illustration), in order to provide stronger evidence for effectiveness of each techniques.

**Questions:**

1. What's the main difference between the "material-aware normal propagation" in the paper and the "normal propagation" in 3DGS-DR? According to 3DGS-DR, their normal propagation are also aware of reflection strength. Is there any key improvement over their solution? Otherwise, is it an adaptation from reflection-strength-aware to PBR-attribute-aware?

---

> ### Author Response · Authors · 2024-11-22
> **Reply to Reviewer 147t**
>
> ## W1: Further qualitative evidence on Inter-reflection technique
>
> We thank the reviewer for raising this question! We have already provided qualitative evidence of the effectiveness of the inter-reflection technique in Figure 9. Regarding the relatively minor improvement in PSNR, this is due to the limited inter-reflection effects present in the Glossy Synthetic dataset. To provide deeper insight into the usefulness of the inter-reflection technique, we have taken the suggestion into account and included visualizations of the indirect lighting components for the Glossy Synthetic dataset in Figure 5 and for the Ref-Real dataset in Figure 12, Appendix of revised manuscript. As illustrated in Figure 12, the real-world scenes feature multiple objects that generate rich inter-reflections.
>
> ## W2: Applying more convincing metrics in ablation study
>
> Thank you for the valuable suggestion! To achieve more accurate normals, we utilize 2D Gaussian primitives (Figure 8), an initial stage with per-Gaussian shading, and a material-aware normal propagation technique (Figure 10). To further demonstrate the impact of each component on our geometry, we have included the mean angular error (MAE) metric for the rendered normal maps on the Shiny Blender dataset [1], as shown in Table 3 in our revised manuscript (note that the Glossy Synthetic dataset [2] does not provide ground truth normal maps).
>
>
> ## Q1: Novelty and effectiveness of material-aware normal propagation
> We apologize for any confusion. The previous normal propagation proposed by 3DGS-DR [3] only considers the connection between reflective strength and the normal accuracy of relative Gaussians, which is often inaccurate. A number of Gaussians are pruned due to their erroneous enlargement. However, Ref-Gaussian employs a more comprehensive physically based rendering equation while assigning BRDF properties to Gaussian, where the experiences of 3DGS-DR are no longer applicable. Techniquely, the accuracy of normls should be associated to the materials. However, positions with inaccurate normals often have difficulty capturing significant specular component due to its sensitivity towards reflected direction. Specifically, for most Gassians in our model, through experiments we confirm the strong positive correlation between normal accuracy and high metallic, low roughness properties(one of the cases where specular component is significant). To that end, we propose material-aware normal propagation, periodically increasing the scale of 2D Gaussians with high metallic and low roughness to propagate their more accurate normal information to adjacent Gaussians, achieving an even better geometry reconstruction quality and faster convergence rate. We have provided further comparisons between previous normal propagation following 3DGS-DR and our material-aware normal propagation in Table 3 and 4, in our revised manuscript, also shown below.
>
> ***Table 3: Comparison of MAE values across components of Ref-Gaussian.***
>
> | **Model**                   | **MAE (avg)** |
> |-----------------------------|---------------|
> | **Ref-Gaussian**                | 2.15 |
> | **w/o 2DGS**                | 4.45          |
> | **w/o Initial stage**       |  3.53 |
> | **w/o Material-aware**      | 3.86          |
> | **w/o PBR**                 | 3.81          |
>
>
>
> **Table 4: Ablation studies on the components of Ref-Gaussian.**
> *w/o Material-aware: use previous normal propagation following 3DGS-DR.*
>
> | **Model**                | **Ref-Gaussian** | **w/o PBR** | **w/o Inter-reflection** |  **w/o Deferred rendering** | **w/o Initial stage** | **w/o Material-aware** |
> |------------------------------|-------------|--------------------------|--------------|----------------------------|-----------------------|-------------------------|
> | **PSNR↑**               | 30.33 | 29.84       |  30.14       | 28.85                      | 29.94                | 29.68                  |
> | **SSIM↑**               | 0.958  | 0.956       |  0.957         | 0.935                      | 0.956                | 0.952                  |
> | **LPIPS↓**              | 0.049  | 0.052       |  0.050         | 0.074                      | 0.052                | 0.056                  |
>
>
>
> ### References
> 1. D. Verbin, P. Hedman, B. Mildenhall, T. Zickler, J. T. Barron, and P. P. Srinivasan. Ref-nerf: Structured viewdependent appearance for neural radiance fields. In CVPR, 2022.
> 2. Y. Liu, P. Wang, C. Lin, X. Long, J. Wang, L. Liu, T. Komura, and W. Wang. Nero: Neural geometry and brdf reconstruction of reflective objects from multiview images. ACM Trans. Graph., 2023.
> 3. K. Ye, Q. Hou, and K. Zhou. 3d gaussian splatting with deferred reflection. In SIGGRAPH, 2024.

---

> > ### Comment · Reviewer_147t · 2024-11-23
> >
> > Thank you to the authors for their thorough clarifications in addressing my concerns regarding Q1 and W2. However, with the qualitative results of Figure 12, I still have a key question regarding the Gaussian-grounded inter-reflection. To my understanding, the inter-reflection modeling in the paper comprises two components: (1) visibility modeling and (2) lighting modeling.
> >
> > 1. **Visibility Modeling.** It appears that this approach is only accurate for specular surfaces, as it produces a deterministic reflected direction, which is noted in L222 and mentioned by other reviewers. Additionally, the approximation from Eq. 8 to Eq. 9 seems to completely ignore the roughness term of BRDF materials, suggesting that the method is designed exclusively for fully specular surfaces.
> > 2. **Lighting Modeling.** This component is represented by Gaussian-level spherical harmonic (SH) attributes. If I understand correctly, these attributes are gradually optimized during training and therefore cannot be reconstructed for relighting purposes, even for specular surfaces, as shown in Figure 12.
> >
> > I believe Fig. 12 illustrates the two limitations outlined above. For (1), the estimated visibility appears binary, creating a strong shadow on the ground, which in turn leads to an overestimated indirect light component on the ground near the two diffuse objects. Also, in the visibility map, the black color (visible=0) of the diffuse object on the left, indicates little contribution of direct lighting to the object's color, which seems incorrect. For (2), if I understand correctly, the scene depicted in Fig. 12 cannot be relit. Otherwise, to address the concern, it may be better to show qualitative relighting results or provide a more detailed decomposition of the scene in Fig. 12 (e.g., diffuse term, direct lighting term).
> >
> > While I might hold my rating, regarding the limitations of the proposed Gaussian-grounded inter-reflection, it is still valuable to mention the impressive normal quality of the method. I think the authors could have highlighted more about the geometry quality, as their ablation study validates the effectiveness of the material-aware normal propagation.

---

> ### Author Response · Authors · 2024-11-25
> **Reply to Reviewer 147t**
>
> Thanks again for the reviewer's valuable time and efforts for reviewing our work, as well as the insightful comments.
>
> **1. Visibility modeling**
>
> Thanks for raising this issue which we should have elaborated and clarified in more details. By design, **our methods excel for both reflective and non-reflective scenes, making it as a unified soluton for modeling a variety of scenes with varying reflection**. This is because, as described in Eq.9, inter-reflection is incorporated just as an additional element into the specular component. Consequently, in less reflective scenes, the influence of inter-reflection diminishes naturally and accordingly as the metallic attribute decreases. In such cases, the diffuse component instead takes over, including the capture of any (usually substantially weak) inter-reflection effect to achieve top performance. This behavior is evident in the decomposition results provided in Figure 5 and Figure 14 (appendix). In contrast, previous reflection focused alternatives such as 3DGS-DR ([1]) are designed specifically only for reflective scenes, leading to narrowed applications. To support this claim, we have included Table 8 in the appendix, which presents a per-scene quantitative comparison among Ref-Gaussian, 3DGS-DR ([1]), and 3DGS ([2]) on the nerf-synthetic dataset, predominantly composed of non-reflective scenes, for novel view synthesis. The results show that our Ref-Raussian still excels over the alternatives, validating its generic and unified advantages.
>
>
> Regarding the roughness term of BRDF materials, we apologize for this confusion and misunderstanding. From Eq.8 to Eq.9, we extend the second term denoted by $L_\mathrm{dir}$ in Eq.8, by introducing the indirect light component $L_\mathrm{ind}$ to capture the inter-reflection effect. There is no approximation in this extension process, nor overlooking of roughness $R$. To improve readability, we have revised Eq.8 and Eq.9 to minimize such confusion in the revised paper.
>
> In a nutshell, we further summarize again how the roughness is involved in our model in case this helps for facilitating a holistic understanding. First, the shadowing-masking term $G$ in $f_s(\omega_i, \omega_o)$, part of BRDF in Eq.7, is a function of the roughness $R$. In the split-sum approximation described in Eq.8, the first term of Eq.8 is precomputed and stored in a 2D lookup texture map with $(\omega_i \cdot N)$ and roughness R as input conditions. During computing the second term of Eq.8, we also utilize roughness for estimating the integral of incident light. To efficiently represent environment lighting, we use trilinear interpolation across a series of pre-integrated cubemaps at varying roughness levels, with the reflected direction and roughness as interpolation parameters.
>
>
> Regarding Figure 12, thanks for the great observation. Per this comment, we find out that in this visualization experiment we mistakenly forgot to apply the normal smooth loss as stated in Section 3.3 (this mistake only applies here but no other experiments), leading to an inaccurate geometry. This causes this seemmingly overestimated indirect light component on the ground near the two diffuse objects. Now we have addressed this issue, and provided the corrected Figure 12 (appendix). As shown in revised visualization, the three components—*diffuse, specular, and indirect light*—obviously complement each other effectively and together produce an excellent final rendering result. We apologies for this mistake.
>
> Regarding the contribution of direct light in the case of previous Figure 12, under this bird eye view, all the lights cast towards that diffuse object will be reflected to the ground, which can be considered as a special object. As a result, its visibility is all zero, meaning no contribution from direct light. Note, *direct light in out context means those whose reflection is not blocked by any scene elements*. We have explicitly explained this in the revised paper.
>
> **References**
> >1. K. Ye, Q. Hou, and K. Zhou. 3d gaussian splatting with deferred reflection. In SIGGRAPH, 2024.
> >2. B. Kerbl, G. Kopanas, T. Leimkühler, and G. Drettakis. *3D Gaussian Splatting for Real-Time Radiance Field Rendering.* ACM Trans. Graph., 2023.

---

> ### Author Response · Authors · 2024-11-25
> **Reply to Reviewer 147t**
>
> **2. Lighting modeling**
>
> Overall, the inter-reflection and relighting components work in a plug-in manner in our approach, but not work simultaneously. We will clarify this further in revision.
> **This flexible design is not our limitation, but a unique advantage over previous alternatives**, which just tackles the reflection challenge alone and is unable to address the relighting problem in a unified framework as our method does. Integrating the two components would make our model more advanced, which we will investigate in the future work.
>
> Please note, we have demonstrated Ref-Gaussian's outstanding relighting performance even without inter-reflection in Figure 11 and the demo video included in the supplementary material. Also note, none of the existing 3DGS-based advancements have allowed for indirect lighting during relighting, remaining an open challenge to be tackled.
>
> **Further illustration**
>
> Additionally, following the reviewer's suggestion to highlight Ref-Gaussian's performance in geometry reconstruction and novel view synthesis, we further provide Figure 13 and Figure 14 (appendix), which further demonstrate our excellent performance by providing more per-scene qualitative results and comparisons of normals with other advanced methods.
>
> Hope our responses clarify the thoughtful questions above, and it would be very much appreciated if the reviewer could kindly check our responses and provide feedback with further questions or concerns (if any). We would be more than happy to address them. Thank you!

---

> > ### Comment · Reviewer_147t · 2024-11-25
> >
> > Thank you for the authors' patient explanation. I truly appreciate the detailed response. I have a few points I’d like to clarify to ensure I haven’t misunderstood.
> >
> > ---
> >
> > ### **Eq. 8 to Eq. 9**
> >
> > In my understanding, for a given scene, there exist ground truth values for $L_\text{dir}$ and $L_\text{ind}$.  Equation 9 is designed to model both direct and indirect lighting components, aiming to approximate these ground truth values as closely as possible. I interpret the transition from Eq. 8 to Eq. 9 as an approximation because the visibility term is estimated through a ray intersection test following a single direction $2(\omega_i\cdot N)N-\omega_i$, which is physically accurate only for fully specular surfaces.
> >
> > However, based on the authors' explanation, it seems Eq. 9 is not intended to be strictly physically accurate (or "relightable"). For example, in Figure 12, it appears the scene would not produce a reasonable relighting result, given such a binary visibility map. Instead, the primary purpose of expanding Eq. 8 to Eq. 9 seems to be minimizing the perturbation caused by occlusion in the environmental lighting estimation during the split-sum process, as depicted in Figure 12.
> >
> > With this understanding, while the novelty of the approach might be diminished (compared to introducing a "relightable" Gaussian-grounded inter-reflection technique), the Ref-Gaussian method achieves state-of-the-art NVS performance for reflective objects. This is a significant accomplishment, and I would consider raising my rating accordingly.

---

> ### Author Response · Authors · 2024-11-26
> **Reply to Reviewer 147t**
>
> Thanks to the reviewer for the insightful feedback and for the time and effort dedicated to thoroughly reviewing our work. We deeply appreciate the thoughtful comments, which have provided valuable perspectives on our research.
>
> We are particularly grateful for the reviewer’s efforts in understanding our method, especially the design and interpretation of our inter-reflection technique. First, we realize the importance of highlight that there are no supervision and ground-truth annotations for $L_\mathrm{ind}$ and $L_\mathrm{dir}$, typically unavailable in practice. Therefore, $L_\mathrm{ind}$ and $L_\mathrm{dir}$ in Eq.9 are all implicitly optimized with the Gaussian premitives while training.
>
> Regarding the understanding of $L_\mathrm{ind}$, we apprciate the view of understanding it as reducing the perturbation caused by occlusion in environmental lighting estimation. This makes sense as occlusion can be considered as the primary cause for the inter-reflection effects. This intention is evident in Figure 12 (appendix) and aligns well with the reviewer’s observations. We have now improved our manuscript thanks to this comment.
>
> Additionally, we sincerely appreciate the recognition of our Ref-Gaussian's strong performance in novel view synthesis.
>
> In our revision, we will ensure these points are explicitly clarified to provide a precise understanding of our approach. The reviewer’s constructive feedback has been instrumental in guiding these refinements, and we are sincerely grateful for this guidance.

---

### Meta-Review · Area_Chair_yMQN · 2024-12-20

**Metareview:**

This paper describes a new Gaussian splatting method called Ref-Gaussian, which aims to achieve real-time high-quality rendering of reflective objects by modeling inter-reflections. The key techniques introduced include geometry enhancement using 2DGS, PBR optimization, and Gaussian-grounded inter-reflection computation. The major strengths are the use of explicit geometry for addressing the inter-reflection issue in Gaussian splatting, and the experimental results demonstrate the effectiveness of the proposed method. On the other hand, the weaknesses was mainly on the lack of evaluation particularly for the inter-reflections and the novelty. The ablation study indicates only a slight decrease in PSNR when rendering without inter-reflections. Regarding the novelty, it was pointed out that many previous 3DGS methods with inverse rendering have adopted similar techniques. Still, it is agreed that the method is carefully designed, and the paper has a merit due to its superior quality compared to existing Gaussian splatting methods. The AC agreed with the reviewers' opinions and rendered this recommendation.

**Additional Comments On Reviewer Discussion:**

There was a concern about the novelty of the proposed method due to the similarity to the existing Gaussian splatting methods that take into account inter-reflections. This point was thoroghly discussed during the author-reviewer discussion phase, and it was agreed that although it is not a significantly new idea, it still has a new component in the visibility modeling scheme and optimization techniques. In addition, it was mentioned that the quality overwhelms the existing ones so it should be shared in the community.

---

### Decision · Program_Chairs · 2025-01-22

Accept (Poster)